# QMP: Q-SWITCH MIXTURE OF POLICIES FOR MULTI-TASK BEHAVIOR SHARING

**Grace Zhang**[1][*]   **Ayush Jain**[1][*]   **Injune Hwang**[2]   **Shao-Hua Sun**[3]   **Joseph J. Lim**[2]
[1]University of Southern California  [2]KAIST  [3]National Taiwan University

## ABSTRACT

Multi-task reinforcement learning (MTRL) aims to learn several tasks simultaneously for better sample efficiency than learning them separately. Traditional methods achieve this by sharing parameters or relabeled data between tasks. In this work, we introduce a new framework for sharing *behavioral policies* across tasks, which can be used in addition to existing MTRL methods. The key idea is to improve each task's off-policy data collection by employing behaviors from other task policies. Selectively sharing helpful behaviors acquired in one task to collect training data for another task can lead to higher-quality trajectories, leading to more sample-efficient MTRL. Thus, we introduce a simple and principled framework called Q-switch mixture of policies (QMP) that selectively shares behavior between different task policies by using the task's Q-function to evaluate and select useful shareable behaviors. We theoretically analyze how QMP improves the sample efficiency of the underlying RL algorithm. Our experiments show that QMP's behavioral policy sharing provides complementary gains over many popular MTRL algorithms and outperforms alternative ways to share behaviors in various manipulation, locomotion, and navigation environments. Videos are available at https://qmp-mtrl.github.io/.

## 1 INTRODUCTION

In multi-task reinforcement learning, each task can benefit from the behaviors learned in others. Consider a robot learning four tasks simultaneously: opening and closing a drawer and a door on a tabletop, as illustrated in Figure 1. A behavior is defined as the policy of how the robot acts in a certain state, with the optimal behavior representing the best response, such as opening its gripper (action) when near the drawer handle (state) in the drawer-open task. As the robot learns, such behaviors are often shareable between tasks. For instance, both drawer-open and drawer-close tasks require behaviors for grasping the handle. Consequently, as the robot refines its ability to grasp the drawer handle in one task, it can incorporate these behaviors into the other, reducing the need to explore the entire

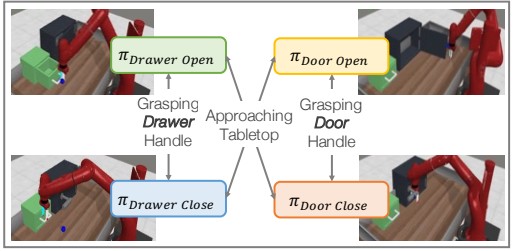

Figure 1: We propose a sample-efficient MTRL framework that selectively shares behaviors by acting with other task policies for data collection. For example, `Drawer Open` and `Drawer Close` can share behaviors performed for grasping drawer handle, while `Drawer Open` and `Door Close` share behaviors for approaching the tabletop.

action space randomly. Following this intuition, can we develop a general framework that leverages such behavior sharing across tasks to accelerate overall learning?

Most multi-task reinforcement learning (MTRL) methods share task information via policy parameters (Vithayathil Varghese & Mahmoud, 2020) or data relabeling (Kaelbling, 1993). We propose a new framework for MTRL: *share behaviors* between tasks to improve data collection by employing potentially useful policies from other tasks for more informative training data. This approach offers a simple, general, and sample-efficient approach that complements existing off-policy MTRL methods.

---

[*]Equal contribution. Correspondence to: {gracez, ayushj}@usc.edu

Prior works (Teh et al., 2017; Ghosh et al., 2018) share behaviors between task policies uniformly by regularizing to one shared distilled policy (Rusu et al., 2015). This introduces a bias towards the mean behavior and causes negative interference when tasks might require differing optimal behaviors from the same state. In contrast, reusing other policies for data collection does not introduce any bias.

We propose *selective behavioral policy sharing* as a novel and general mechanism to improve sample efficiency in any MTRL architecture. Our key insight is that behaviors being acquired in other tasks can help when appropriately selected and shared, as shown in human learners (Tomov et al., 2021). In the Drawer Open task, while learning to approach the drawer handle, the robot should share behaviors between the Drawer policies, but avoid Door policies which would lead it to the wrong object.

The key question with selective behavioral policy sharing is how to identify helpful behaviors from other policies in a principled way. We propose a principled way of selecting shared behaviors: a Q-switch Mixture of Policies (QMP). At each state, one policy from a mixture of all policies is selected to collect data. The Q-switch makes this selection based on which policy best optimizes the current task's soft Q-value because that is an estimate of the most helpful behavior for the current task. We prove that this selection mechanism preserves the convergence guarantees of the underlying RL algorithm and potentially improves sample efficiency. Crucially, QMP uses other tasks' policies only for data collection, allowing policy training to remain unbiased under any off-policy RL algorithm.

Our primary contribution is introducing behavioral policy sharing for MTRL as a novel avenue of information sharing between tasks and addressing the problem of principled selective behavior sharing. Our proposed framework, Q-switch Mixture of Policies (QMP), can effectively identify shareable behaviors between tasks and incorporates them to gather more informative training data for off-policy RL. We prove that QMP's behavior sharing not only preserves the policy convergence of the underlying RL algorithm, but is at least as sample efficient. We demonstrate that QMP provides complementary gains to other forms of MTRL in a range of manipulation, locomotion, and navigation tasks and performs well over diverse task families when compared to other behavior sharing methods.

## 2 RELATED WORK

**Information Sharing in Multi-Task RL.** There are multiple, mostly complementary ways to share information in MTRL, including sharing *data*, sharing *parameters* or *representations*, and sharing *behaviors*. In offline MTRL, prior works selectively share *data* between tasks (Yu et al., 2021; 2022). Sharing parameters across policies can speed up MTRL through shared *representations* (Xu et al., 2020; D'Eramo et al., 2020; Yang et al., 2020; Sodhani et al., 2021; Misra et al., 2016; Perez et al., 2018; Devin et al., 2017; Vuorio et al., 2019; Rosenbaum et al., 2019; Yu et al., 2023; Cheng et al., 2023; Hong et al., 2022) and can be easily combined with other types of information sharing. Most similar to our work, Teh et al. (2017) and Ghosh et al. (2018) share *behaviors* between multiple policies through policy distillation and regularization. Vuong et al. (2019) identify which states between tasks share optimal behavior and regularize to each other there. These works share behaviors through regularization, biasing the policy objective when tasks have differing optimal behaviors. In contrast, our work selectively shares behavioral policies without modifying the training objective.

**Multi-Task Learning for Diverse Task Families.** Multi-task learning in diverse task families is susceptible to *negative transfer* between dissimilar tasks, hindering training. Prior works combat this by measuring task relatedness through validation loss on tasks (Liu et al., 2022; Ackermann et al., 2021) or influence of one task to another (Fifty et al., 2021; Standley et al., 2020) to find task groupings for training. Other works focus on the challenge of multi-objective optimization (Sener & Koltun, 2018; Hessel et al., 2019; Yu et al., 2020; Liu et al., 2021; Schaul et al., 2019; Chen et al., 2018; Kurin et al., 2022). Similar to these works, we identify that prior behavior-sharing MTRL approaches are susceptible to negative transfer. However, we avoid the challenge of negative transfer entirely by selectively sharing behaviors only during off-policy data collection.

**Exploration in Multi-Task Reinforcement Learning.** Our approach of modifying the behavioral policy to leverage shared task structures can be seen as a form of MTRL exploration, which we discuss further in Appendix Section 21c. Bangaru et al. (2016) encourage agents to increase their state coverage by providing an exploration bonus. Zhang & Wang (2021) study sharing information between agents to encourage exploration under tabular MDPs. Kalashnikov et al. (2021b) directly leverage data from policies of other specialized tasks (like grasping a ball) for their general task

variant (like grasping an object). In contrast to these approaches, we do not require a pre-defined task similarity measure or exploration bonus; we demonstrate in Section 6 that QMP works across many tasks and domains without these additional measures. Skill learning can be seen as behavior sharing in a single task setting such as learning options for exploration or heirarchical RL (Machado et al., 2017; Jinnai et al., 2019b;a; Hansen et al., 2019; Riemer et al., 2018). We also discuss the difference to single-task exploration in Appendix Section G.3.

**Using Q-functions as filters.** Yu et al. (2021) uses Q-functions to filter which data should be shared between tasks in a multi-task setting. In the imitation learning setting, Nair et al. (2018) and Sasaki & Yamashina (2020) use Q-functions to filter out low-quality demonstrations, so they are not used for training. In both cases, the Q-function is used to evaluate some data that can be used for training. Zhang et al. (2022) reuses pre-trained policies to learn a new task, using a Q-function as a filter to choose which pre-trained policies to regularize to as guidance. In contrast to prior works, our method uses a Q-function to *evaluate* different task policies to gather training data.

## 3 PROBLEM FORMULATION

Multi-task reinforcement learning (MTRL) addresses sequential decision-making tasks, where an agent learns a policy to act optimally in an environment (Kaelbling et al., 1996; Wilson et al., 2007). Therefore, in addition to typical multi-task learning techniques, MTRL can also share *behaviors*, i.e., actions, to improve sample efficiency. However, current approaches share behaviors uniformly (Section 2), which assumes that different tasks' behaviors do not conflict. To address this limitation, we seek to develop a selective behavior-sharing method that can be applied in more general task families for sample-efficient MTRL.

**Multi-Task RL with Behavior Sharing.** We aim to simultaneously learn a set $\{\mathbb{T}_1, \ldots, \mathbb{T}_N\}$ of $N$ tasks. Each task $\mathbb{T}_i$ is a Markov Decision Process (MDP) defined by state space $\mathcal{S}$, action space $\mathcal{A}$, transition probabilities $\mathcal{T}_i$, reward functions $\mathcal{R}_i$, initial state distribution $\rho_i$, and discount factor $\gamma \in [0, 1]$. While we use $\mathcal{S}$ to denote shared state spaces for simplicity, our formulation extends to tasks with different state spaces as it complements policy architectures that share state encoders. The agent learns a set of $N$ policies $\{\pi_1, \ldots, \pi_N\}$, where each policy $\pi_i(a|s)$ represents the behavior on task $\mathbb{T}_i$. The objective is to maximize the average expected return over all tasks,

$$\{\pi_1^*, \ldots, \pi_N^*\} = \max_{\{\pi_1, \ldots, \pi_N\}} \frac{1}{N} \sum_{i=1}^{N} \left[ \mathbb{E}_{a_t \sim \pi_i} \sum_{t=0}^{\infty} \gamma^t \mathcal{R}_i(s_t, a_t) \right].$$

Unlike prior works, our tasks can exhibit conflicting optimal behaviors: for any $s$, $\pi_i^*(a|s)$ may differ from $\pi_j^*(a|s)$. Thus, prior methods that bias policy learning objectives like direct policy sharing (Kalashnikov et al., 2021a) or behavior regularization (Teh et al., 2017) would be suboptimal.

## 4 APPROACH

To improve the sample efficiency of multi-task RL, we propose a framework that *selectively* incorporates behaviors from policies of other tasks without introducing bias into the RL objective for the current task. We achieve this by using a mixture of all policies as the behavioral policy for the current task, thereby modifying only its *off-policy training data*. However, naively mixing other policies into the current task's behavioral policy does not necessarily improve its sample efficiency. To address this, we derive a specific definition of this mixture, named Q-switch Mixture of Policies (QMP), that selects a policy based on the current task's Q-function (see Figure 2 and Algorithm 1) and prove that QMP guarantees greater than or equal sample efficiency than using the current task's policy alone.

### 4.1 MULTI-TASK BEHAVIOR SHARING VIA OFF-POLICY DATA COLLECTION

MTRL methods like Teh et al. (2017) use *regularization to a common average policy* to enforce task policies to share behaviors. However, this introduces bias to each policy's RL objective, leading to suboptimal actions in states where tasks require different actions. To address this, we propose using a *mixture of policies for off-policy data collection* as the means of behavior-sharing. At each state in any given task, one of the task policies is selected to gather training data as the current behavioral policy. This approach is compatible with any off-policy RL algorithm (Watkins & Dayan, 1992)

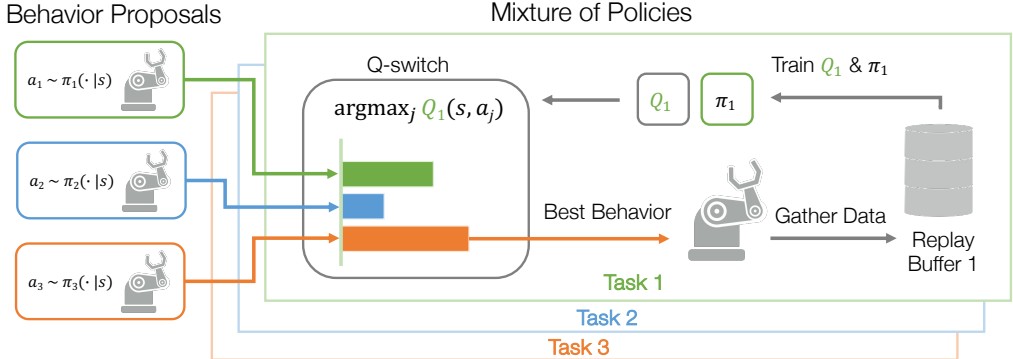

Figure 2: Our method (QMP) shares behavior between task policies in the data collection phase using a mixture of these policies. For example, in Task 1, each task policy proposes an action $a_j$. The task-specific Q-switch evaluates each $Q_1(s, a_j)$ and selects the best scored policy to gather reward-labeled data to train $Q_1$ and $\pi_1$. Thus, Task 1 will be boosted by incorporating high-reward shareable behaviors into $\pi_1$ and improving $Q_1$ for subsequent Q-switch evaluations.

because the environment rewards help determine the best actions from the collected data. However, an effective mixture policy must choose the behavioral policies in a selective and principled way.

**Definition 4.1** (Mixture of Policies). For each task $\mathbb{T}_i$, the mixture policy $\pi_i^{\text{mix}}(a \mid s)$ is defined as $\pi_i^{\text{mix}}(a \mid s) = \pi_{f_i\left(s, \pi_1, \ldots, \pi_N\right)}(a \mid s)$, where $f_i(s, \pi_1, \ldots, \pi_N) : \mathcal{S} \times \Pi^N \to \{1, \ldots, N\}$ is a mixture-switch function that selects one of the policies $\pi_1, \ldots, \pi_N$ based on the current state $s$.

Our intuition of policy mixture shares inspiration with hierarchical RL (Çelik et al., 2021; Daniel et al., 2016; End et al., 2017; Goyal et al., 2019) where a *mixture* of options is learned according to the downstream task(s). However, a key difference in an MTRL mixture is that each policy is optimized for its own specific task and not designed to fit the task where the mixture is employed.

### 4.2 Q-SWITCH MIXTURE OF POLICIES (QMP)

We aim to derive a principled mixture-switch function $f_i$ such that the mixture policy $\pi_i^{\text{mix}}$ selectively incorporates behaviors from other policies while being guaranteed to be no worse than the current task's policy $\pi_i$. We recall the generalized policy iteration procedure (Sutton & Barto, 2018) underlying single-task SAC (Haarnoja et al., 2018): policy evaluation learns $Q$ by minimizing the bellman error on the collected data, and policy improvement follows $Q$ by minimizing the KL divergence between the new policy and the exponential of the current $Q$-function, $Q^{\pi^{\text{old}}}$:

$$\pi^{\text{new}} = \arg\min_{\pi' \in \Pi} D_{\text{KL}}\left(\pi'(\cdot \mid s_t) \middle\| \frac{\exp\left(\frac{1}{\alpha}Q^{\pi^{\text{old}}}(s_t, \cdot)\right)}{Z^{\pi^{\text{old}}}(s_t)}\right) \tag{1}$$

In practice, the gradient updates in SAC are gradual and do not instantly achieve this optimization in Eq. 1, leaving a suboptimality gap to catch up to the Q-function. Thus, a mixture policy $\pi_i^{\text{mix}}$ that selects the best policy from a set of all given policy candidates, *including the current policy*, ensures that $\pi_i^{\text{mix}}$ is at least as good as $\pi_i$ for the current state $s$, while potentially being a better optimizer of Eq. 1 due to shareable behaviors from the other task policies:

$$\min_{\pi' \in \{\pi_1, \ldots, \pi_N\}} D_{\text{KL}}\left(\pi'(\cdot \mid s) \middle\| \frac{\exp(\frac{1}{\alpha}Q^{\pi_i}(s, \cdot))}{Z^{\pi_i}(s)}\right) \leq D_{\text{KL}}\left(\pi_i(\cdot \mid s_t) \middle\| \frac{\exp(\frac{1}{\alpha}Q^{\pi_i}(s, \cdot))}{Z^{\pi_i}(s)}\right) \tag{2}$$

Simplifying the expression on the left results in the following definition (derivation in Appendix B).

**Definition 4.2** (Q-switch Mixture of Policies: QMP). For a task $\mathbb{T}_i$ and available candidate policies $\{\pi_1, \ldots, \pi_N\}$, the QMP $\pi_i^{\text{mix}}(a \mid s)$ selects a policy according to:

$$\pi_i^{\text{mix}} = \arg\max_{\pi' \in \{\pi_1, \ldots, \pi_N\}} \mathbb{E}_{a \sim \pi'(\cdot \mid s)}\left[Q^{\pi_i}(s, a)\right] + \alpha\mathcal{H}\left[\pi'(\cdot \mid s)\right] \tag{3}$$

Algorithm 1 shows that QMP can be plugged into any MTRL framework, making it complementary with various MTRL frameworks like parameter-sharing and data relabeling (see Section 7.1). In practice, we estimate the expectation in Eq. 3 by evaluating the Q-value for the mean action of each task policy's distribution $\pi'(\cdot|s)$ ignoring the entropy term. We do not find any empirical difference when using a sampled estimate of the expectation (see Appendix G.2) or including the entropy term, as the Q-value is the primary distinguishing factor between policies. In terms of compute, sampling from QMP's $\pi_i^{\mathrm{mix}}(a|s)$ does require more policy and Q-function evaluations. However, evaluations are parallelized and impact runtime negligibly, as shown in Appendix G.4.

---

**Algorithm 1** Q-switch Mixture of Policies (QMP)

**Input:** Task Set $\{\mathbb{T}_1, \ldots, \mathbb{T}_N\}$
Initialize $\{\pi_i\}_{i=1}^N$, $\{Q_i\}_{i=1}^N$, Data buffers $\{\mathcal{D}_i\}_{i=1}^N$
**for** each epoch **do**
    **for** $i = 1$ to $N$ **do**
        **while** Task $\mathbb{T}_i$ episode not terminated **do**
            Observe state $\mathbf{s}$
            Compute $\pi_i^{\mathrm{mix}}$ as in Eq. 3.
            Take action proposal from $a \sim \pi_i^{\mathrm{mix}}$
            $\mathcal{D}_i \leftarrow \mathcal{D}_i \cup (s, a, r_i, s')$
        **end while**
    **end for**
    **for** $i = 1$ to $N$ **do**
        Update $\pi_i, Q_i$ using $\mathcal{D}_i$ with SAC
    **end for**
**end for**
**Output:** Trained policies $\{\pi_i\}_{i=1}^N$

---

While $\pi_i^{\mathrm{mix}}$ can mistakenly choose a poor policy due to estimation error in $Q^{\pi_i}$, this is identical to Q-learning or SAC, where the Q-function would be inaccurate at less-seen states. In both Q-learning and QMP, this is corrected with online interactions where the Q-function is trained to be more accurate in a subsequent iteration. Furthermore, $\pi_i^{\mathrm{mix}}$ actually better maximizes $Q^{\pi_i}$ than $\pi_i$, which is the objective under generalized policy iteration. Note that QMP does **not** exacerbate the problem of overestimation because the soft policy evaluation step stays the same, i.e., it uses $\pi_i$ and not $\pi_i^{\mathrm{mix}}$.

## 5 Why QMP Works: Theory and Didactic Example

### 5.1 QMP: Convergence and Improvement Guarantees

QMP performs simultaneous MTRL by collecting data using a Q-switch guided mixture of policies $\pi_i^{\mathrm{mix}}$. In Appendix C, we prove that QMP with underlying RL algorithm Soft-Actor Critic (SAC) (Haarnoja et al., 2018) shares the same convergence guarantees in a tabular setting. Particularly, we show that under QMP, policy evaluation converges because QMP only modifies data collection of off-policy RL, policy improvement guarantees are preserved (Theorem 5.1), and policy iteration converges to an optimal policy at least as sample-efficiently (Theorem C.2).

The key reason for *better policy improvement* of QMP over the current task policy $\pi_i$ is the $\arg\max$ operation in Eq. 3, which ensures that the selected policy $\pi_i^{\mathrm{mix}} \in \{\pi_j\}_{j=1}^N$ optimizes the SAC objective at least as well as $\pi_i$ itself. We formalize this in Theorem 5.1 with proof in Appendix C.1. Due to the suboptimality gap in Eq. 1 in SAC, QMP can actually achieve better policy improvement when there are shareable behaviors between policies.

**Theorem 5.1** (Mixture Soft Policy Improvement). *Consider $\pi_i^{old}$ and its associated Q-function $Q_i$. Apply SAC's policy improvement $\pi_i^{old} \rightarrow \pi_i$ and then $\pi_i \rightarrow \pi_i^{mix}$ from Eq. 3. Then, $Q^{\pi_i^{mix}}(\mathbf{s}_t, \mathbf{a}_t) \geq Q^{\pi_i}(\mathbf{s}_t, \mathbf{a}_t) \geq Q^{\pi_i^{old}}(\mathbf{s}_t, \mathbf{a}_t)$ for all tasks $\mathbb{T}_i$ and for all $(s_t, a_t) \in \mathcal{S} \times \mathcal{A}$ with $|\mathcal{A}| < \infty$.*

While QMP in Def. 4.2 applies to any set of candidate policies $\{\pi_1, ..., \pi_N\}$, one expects $\pi_i^{\mathrm{mix}}$ to improve over $\pi_i$ when some $\pi_j \neq \pi_i$ proposes an action candidate better than $\pi_i$ for Task $\mathbb{T}_i$. This is more likely in MTRL policies that *share structure between tasks* than an arbitrary set of policies. For instance, if $\mathbb{T}_i$ and $\mathbb{T}_j$ share a subtask that appears early in the episodes for $\mathbb{T}_j$, then $\pi_j$ would have already learned it before $\pi_i$ and be a better policy for certain states of $\mathbb{T}_i$, according to $Q_i$.

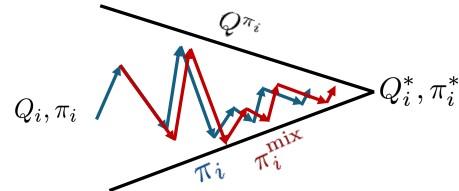

Figure 3: QMP generalized policy iteration

QMP making bigger policy improvement steps results in each iteration of generalized policy iteration making more progress towards optimality. This reduces the number of iterations required to converge, improving the sample efficiency of the algorithm as illustrated in Fig. 3 and proved in Theorem C.2.

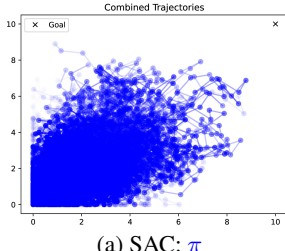
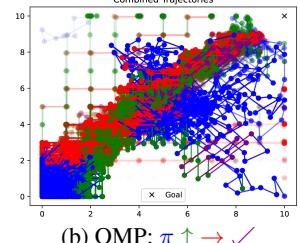
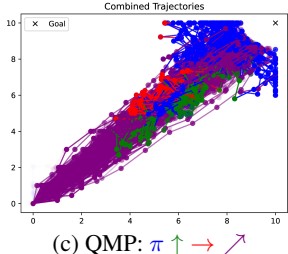

(a) SAC: $\pi$      (b) QMP: $\pi \uparrow \rightarrow \swarrow$      (c) QMP: $\pi \uparrow \rightarrow \nearrow$

Figure 4: **2D Point Reaching.** We visualize the training trajectories of $\pi$ with different sets of task policies (fixed but stochastic) and color each step with the policy that proposed it. **(a)** The single-task SAC policy cannot reach the goal. **(b)** With 3 diverse policies ($\uparrow \rightarrow \swarrow$), QMP often selects other policies, showing the suboptimality gap from $Q$ in Eq. 1. **(c)** When a highly relevant $\nearrow$ policy is added, QMP often selects $\nearrow$ as it is likely to best optimize the learned Q-function.

## 5.2 ILLUSTRATIVE EXAMPLE: 2D POINT REACHING

We demonstrate when QMP can *utilize alternate policy candidates* $\{\pi_1, \ldots, \pi_N\}$ to more effectively learn a policy by bridging a *policy improvement suboptimality gap* as $\pi$ tries to follow $Q$ in Eq. 1. Consider a 2D point-reaching task where the agent must navigate from the bottom-left corner $(0, 0)$ to the goal in the top-right corner $(10, 10)$. The point agent receives dense rewards based on its proximity to the goal and takes incremental 2D actions $(\Delta x, \Delta y) \in [-1, 1]^2$.

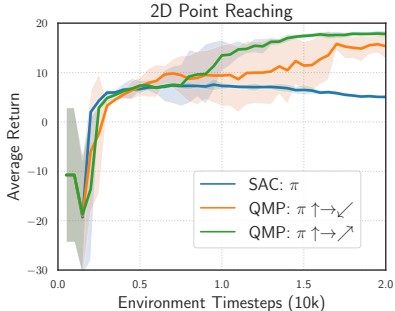

Figure 5: QMP improves performance using other policies, increasingly so when they are task-relevant (5 seeds).

Figure 5 shows that the SAC policy $\pi$ converges to a suboptimal solution. Fig. 4a confirms that the data collected by SAC policy never reaches the goal. This shows that if the suboptimality gap in $\pi$ is not successfully bridged, it can make the entire algorithm converge suboptimally.

To illustrate the effect of QMP, we add 3 fixed gaussian policies centered on ($\uparrow \rightarrow \swarrow$) or ($\uparrow \rightarrow \nearrow$), and only let $\pi$ be trainable. Fig. 4b, 4c show that $\pi_i^{\text{mix}}$ employs alternate policies at many states in data collection as they optimize Eq. 3 better than $\pi$ itself. This *selectivity* enables $\pi_i^{\text{mix}}$ to generate more effective goal-reaching trajectories by bridging the suboptimality gap, resulting in better performance in Fig. 5. A policy like $\nearrow$ that is more relevant to the underlying task leads to a larger gain.

The same principle extends to the simultaneous multi-task RL setting. In MTRL, each task's policy continuously improves and can serve as a valuable candidate in the mixture for other tasks. QMP enables tasks to selectively share their behaviors, allowing each task to benefit from the progress of others. This mutual assistance accelerates learning across all tasks, as the mixture policy $\pi_i^{\text{mix}}$ for each task $\mathbb{T}_i$ selects the most promising action proposals from all available policies according to the task-specific Q-function, guaranteed to be at least as good as $\pi_i$ itself. Consequently, MTRL combined with QMP leverages the collective knowledge of all tasks to bridge suboptimality gaps more efficiently, leading to improved sample efficiency and overall performance.

## 6 EXPERIMENTS

### 6.1 ENVIRONMENTS

We evaluate our method in 7 multi-task designs in manipulation, navigation, and locomotion environments, shown in Figure 6. These tasks vary in the degree of shared and conflicting behaviors between tasks and the number of tasks in the set. Further details in Appendix Section D.

**Multistage Reacher:** A 6 DoF Jaco arm learns 5 tasks with ordered subgoals. The first 4 tasks share some subgoals, while the 5th *conflicting* task requires the agent to stay at its initial position.

**Maze Navigation:** Building on point mass maze navigation (Fu et al., 2020), we define 10 tasks with various start and goal locations exhibiting coinciding and conflicting segments in the optimal paths.

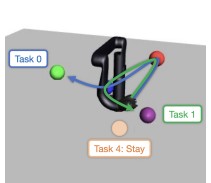 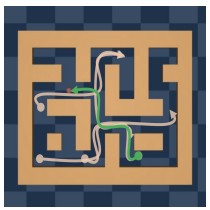 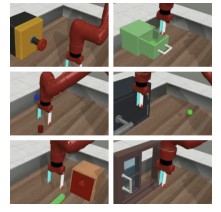 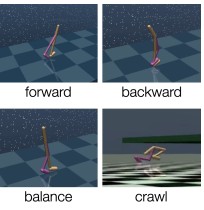 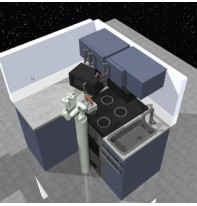

(a) Jaco Reacher  (b) Maze Navigation  (c) Meta-World  (d) Walker2D  (e) Franka Kitchen

Figure 6: **Environments & Tasks**: (a) Multistage Jaco Reacher. The agent must reach different subgoals or stay still (Task 4). (b) Maze Navigation. The agent (green circle) must navigate to the goal (red circle). 4 other tasks are shown in orange. (c) Meta-World: 10 table-top manipulation tasks. (e) Franka Kitchen: 10 tasks, interacting with one appliance or cabinet.

**Meta-World Manipulation:** We use three task sets based on the Meta-World environment (Yu et al., 2019). **Meta-World MT10** and **Meta-World MT50** are sets of 10 and 50 table-top manipulation tasks involving different objects and behaviors. **Meta-World CDS** is a 4-task setup proposed in Yu et al. (2021) which places the door and drawer objects next to each other on the same tabletop so that all 4 tasks (door open & close, drawer open & close) are solvable in a simultaneous multi-task setup.

**Walker2D:** Walker2D is a 9 DoF bipedal walker agent with the multi-task set containing 4 locomotion tasks proposed in Lee et al. (2019): walking forward, walking backward, balancing, and crawling. These tasks require different gaits without an obviously identifiable shared behavior in the optimal policies but can still benefit from intermediate behaviors like balancing.

**Kitchen:** We use the challenging manipulation environment proposed by Gupta et al. (2019) where a 9 DoF Franka robot performs tasks in a kitchen. We create a task set out of 10 manipulation tasks: turning on or off different burners and light switches, and opening or closing different cabinets.

## 6.2 BASELINES

We first select popular and representative MTRL methods that share other forms of information to evaluate how behavior-sharing with QMP improves their performance:

- **No-Sharing** consists of $N$ (refers to number of tasks) independent RL architectures where each agent is assigned one task and trained to solve it without any information sharing with other agents.

- **Data-Sharing (UDS)** proposed in Yu et al. (2022) shares data between tasks, relabelling with minimum task reward. We modified this offline RL algorithm to online.

- **Parameter-Sharing** a multi-head SAC policy sharing parameters but not behaviors over tasks.

We validate QMP's approach to share behaviors via *off-policy data collection* with other approaches:

- **No-Shared-Behaviors** consists of $N$ RL agents where each agent is assigned one task and trained to solve it without any behavior sharing with other agents: no bias and no sharing.

- **Fully-Shared-Behaviors** is a single SAC agent that learns one shared policy for all tasks, outputting the same action for a given state regardless of task (full parameter sharing): fully biased sharing.

- **Divide-and-Conquer RL (DnC)** (Ghosh et al. (2018)) uses $N$ policies that share behaviors through policy distillation and regularization to the mean (adapted for MTRL): biased objective for sharing.

- **DnC (Regularization Only)** is a no policy distillation variant of DnC we propose as a baseline.

- **QMP (Ours)** learns $N$ policies that share behaviors in off-policy data collection: unbiased sharing.

Our code is available here https://github.com/clvrai/qmp. We used SAC Haarnoja et al. (2018) for all environments and methods. All the non-parameter sharing baselines use the same SAC hyperparameters. Please refer to Appendix H for complete details.

## 7 RESULTS

Our experiments address: (1) Does QMP provide complementary gains to other forms of MTRL? (2) How does sharing behavioral policies compare with alternate forms of behavior sharing? (3) Can QMP effectively identify shareable behaviors? (4) Ablating key components of QMP.

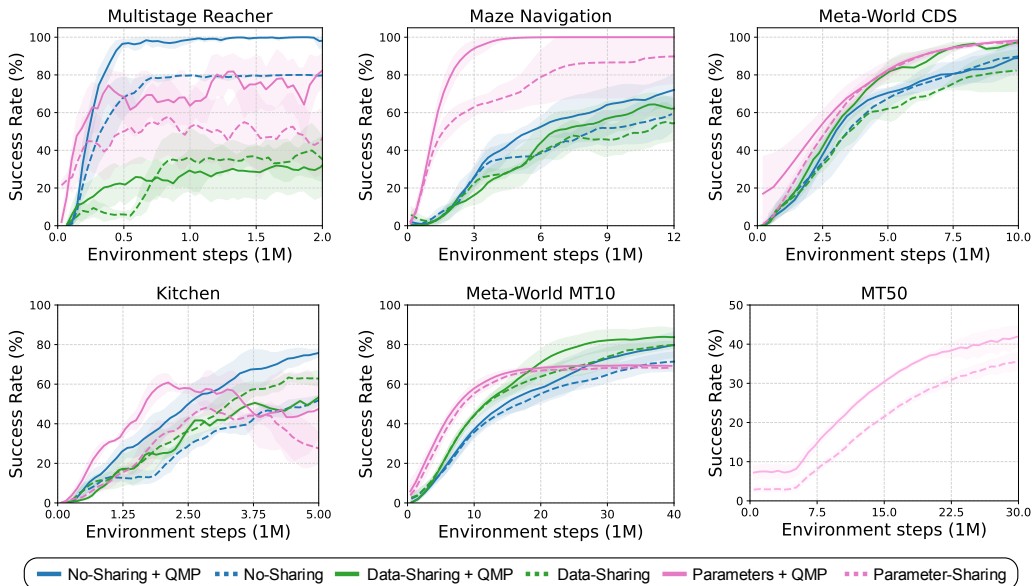

Figure 7: **Behavioral policy sharing is complementary**. QMP (solid lines) shows general improvement over MTRL frameworks (same-colored dashed lines) like No shared architecture (blue), shared parameters (pink), and shared data (green). Methods without parameter-sharing on MT50 converge very slowly. Success rate means and std (shaded) are over $N$ tasks, 10 episodes per task, and 5 seeds.

## 7.1 IS BEHAVIOR SHARING COMPLEMENTARY TO OTHER MTRL FRAMEWORKS?

We demonstrate that our method is compatible with and provides complementary performance gains with other forms of MTRL that share different kinds of information, including parameter sharing and data sharing. We compare the performance between 3 base MTRL algorithms, No-Sharing, Parameter-Sharing, and Data-Sharing, with the addition of QMP in Figure 7. The No-Sharing baseline provides a baseline comparison of QMP's effectiveness on its own. For the Parameter-Sharing and Data-Sharing baselines we chose the base algorithms for their popularity and simplicity. In each case, we add QMP by simply replacing the data collection policy with $\pi_i^{\text{mix}}$. We find that **QMP is complementary to all three baseline frameworks**, mostly with additive performance gains in sample efficiency and final performance, while not hurting the performance of the base method in all but one case (Data-Sharing in Kitchen). We additionally see that QMP improves PCGrad's (Yu et al., 2020) performance significantly in 3 out of 4 environments tested in Appendix E.4. This shows that QMP is a simple and complementary addition to other forms of MTRL.

QMP significantly improves upon the No-Sharing baseline in all environments except Meta-World CDS where it performs comparatively. This demonstrates that sharing behavioral policies is a promising avenue for efficient and performant MTRL. In the data-sharing comparison, we see that the addition of QMP improves or performs comparatively to the base algorithm. In Multistage Reacher and Maze Navigation, we see that both Data-Sharing and Data + QMP perform worse than the other MTRL methods, highlighting the fact that sharing data directly between tasks can be ineffective without access to a re-labeled task rewards like in our setting. In environments where data-sharing does well, like Meta-World CDS, we see that adding QMP does improve sample efficiency.

We find that Parameters + QMP generally outperforms Parameter-Sharing, while inheriting its sample efficiency gains. In many cases, the parameter-sharing methods converge sub-optimally, highlighting that shared parameter MTRL has its own challenges. However, in Maze Navigation, we find that sharing **Parameters + Behaviors greatly improves the performance over both the Parameter-Sharing baseline *and* No-Sharing + QMP variant of QMP**. This demonstrates the additive effect of these two forms of information sharing in MTRL. The agent initially benefits from the sample efficiency gains of the multi-head parameter-sharing architecture, while behavior sharing accelerates learning by selectively using other policies to keep learning even after the parameter-sharing effect plateaus. demonstrating the compatibility between QMP and parameter sharing as key ingredients to sample efficient MTRL. We further highlight that this **benefit of QMP increases with the number of tasks** increasing from 10 to 50 in Meta-World, where we see that QMP is actually more effective

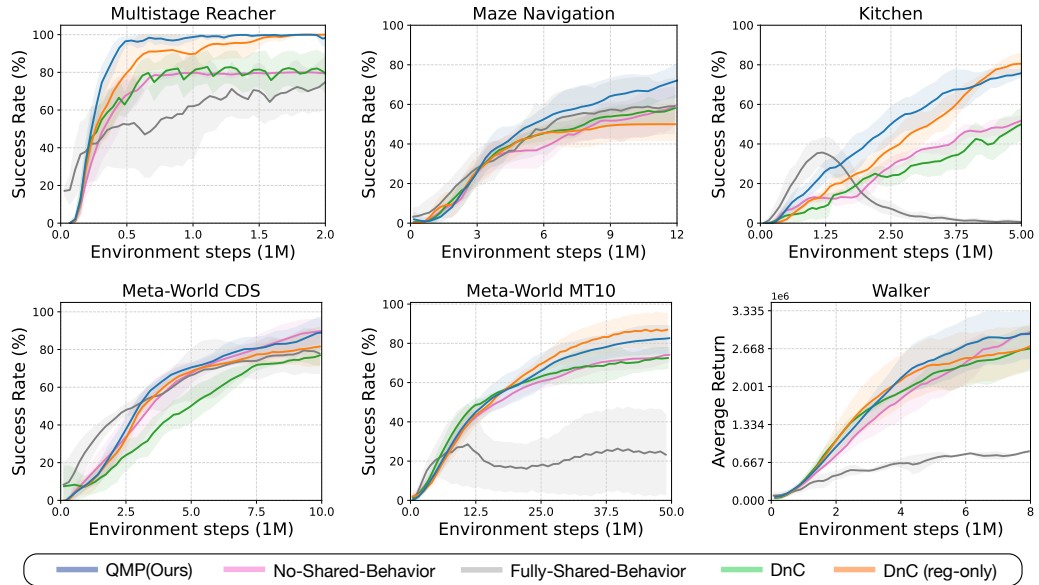

Figure 8: **QMP reliably shares behaviors**. In task sets exhibiting conflicting behaviors, QMP consistently matches or exceeds baselines in rate of convergence and final performance.

when combined with parameter sharing in MT50 than in MT10. QMP scales well with the number of tasks and can actually perform better likely due to *more shared behaviors* in the larger task set.

## 7.2 BASELINES: COMPARING DIFFERENT APPROACHES TO SHARE BEHAVIORS

To verify QMP's efficacy as a behavior-sharing mechanism, we evaluate baselines that share behaviors in different ways on 6 environments in Figure 8. QMP reliably matches or exceeds other methods, especially in tasks that require conflicting behaviors, where alternate approaches are ineffective.

In the task sets with the most directly conflicting behaviors, Multistage Reacher and Maze Navigation, our method clearly outperforms other behavior-sharing and data-sharing baselines. In Multistage Reacher, our method reaches > 90% success rate at 0.5 million environment steps, while DnC (reg.), the next best method, takes 3 times the number of steps to fully converge. The rest of the methods fail to attain the maximum success rate. We also see that QMP scales better from 3 to 10 tasks in Maze compared to other behavior sharing methods in Appendix Section E.2.

In the remaining task sets with no directly conflicting behaviors, we see that QMP is competitive with the best-performing baseline for more complex manipulation and locomotion tasks. Particularly, in Walker2D and Meta-World CDS, we see that QMP is the most sample-efficient method and converges to a better final performance than any other behavior sharing method. In Meta-World MT10 and Kitchen, DnC (regularization only) also performed very well, showing that well-tuned uniform behavior sharing can be very effective in tasks without conflicting behavior. However, QMP also performs competitively and more sample efficiently, showing that QMP is effective under the same assumptions as uniform behavior-sharing methods but can do so *adaptively* and across more *general task families*. The Fully-Shared-Behaviors baseline often performs poorly because it totally biases the policies, while the No-Shared-Behavior is a surprisingly strong baseline as it introduces no bias.

## 7.3 CAN QMP EFFECTIVELY IDENTIFY SHAREABLE BEHAVIORS?

Figure 9a shows the average proportion of sharing from other tasks for Multistage Reacher Task 0 over the course of training. We see that QMP learns to generally share less behavior from Policy 4 than from Policies 1-3 (Appendix Figure 20). Conversely, QMP in Task 4 also shares the least total cross-task behavior (Appendix Figure 19). We see this same trend across all 5 Multistage Reacher tasks, showing that the Q-switch successfully **identifies conflicting behaviors that should not be shared**. Further, Figure 9a also shows that **total behavior-sharing from other tasks goes down over training**. Thus, Q-switch learns to prefer its own task-specific policy as it becomes more proficient.

We qualitatively analyze how behavior sharing varies within a single episode by visualizing a QMP rollout during training for the Drawer Open task in Meta-World CDS (Figure 9b). We see that it makes reasonable policy choices by (i) switching between all 4 task policies as it approaches the

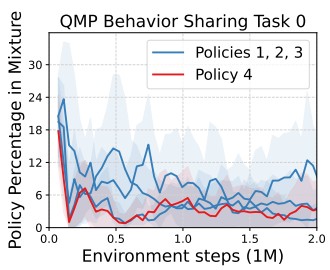

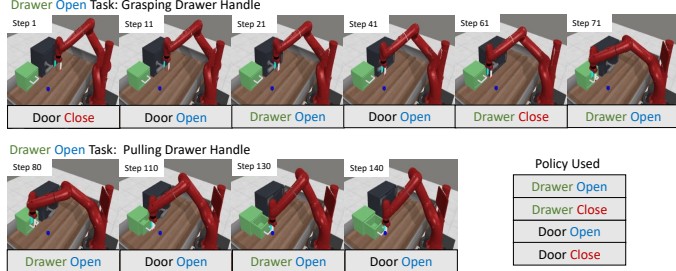

(a) Behavior-sharing over training

(b) Behavior-sharing in a single training episode.

Figure 9: (a) Mixture probabilities of other policies for Task 0 in Multistage Reacher with the conflicting task Policy 4 shown in red. (b) Policies chosen by the QMP behavioral policy every 10 timesteps for the Drawer Open task throughout one training episode. The policy approaches and grasps the handle (top row), then pulls the drawer open (bottom row).

drawer (top row), (ii) using drawer-specific policies as it grasps the drawer-handle, and (iii) using Drawer Open and Door Open policies as it pulls the drawer open (bottom row). In conjunction with the overall results, this supports our claim that QMP can effectively identify shareable behaviors between tasks. For details on this visualization and the full analysis results see Appendix Section F.

Inspired by hierarchical RL (Dabney et al., 2021) and multi-task exploration (Xu et al., 2024), we briefly investigate *temporally extended behavior sharing* in Appendix E.6. Recently, Xu et al. (2024) showed that if one assumes a high overlap between optimal policies of different tasks, other task policies can aid exploration. So, we simply roll out each policy QMP selects for a fixed number of steps. QMP theory no longer holds as it requires selecting a policy at every step. Yet, this naive temporally extended QMP yields improvements in **some** environments like Maze with strong overlap.

## 7.4 ABLATIONS

We show the importance of Q-switch in QMP (Def. 4.2) against alternate forms of policy mixtures (Def. 4.1). **QMP-Uniform** is a uniform distribution over policies, $f_i = \mathbb{U}(\{1, \ldots, N\})$ and achieves only 60% success rate (Figure 10), showing the importance of selectivity. **QMP-Domain-Knowledge** is a hand-crafted, fixed policy distribution based on an estimate of similarity between tasks. Multistage Reacher measures this similarity by the shared sub-goal sequences between tasks (Appendix D). QMP-Domain performs well initially but plateaus early, showing that which behaviors are shareable depends on the state and current policy. We further ablate the $\arg\max$ in Q-switch against a softmax variation resulting in a *probabilistic mixture policy* in Appendix Section G.1, and evaluating on the *mean policy actions* (Appendix Section G.2) to validate our design.

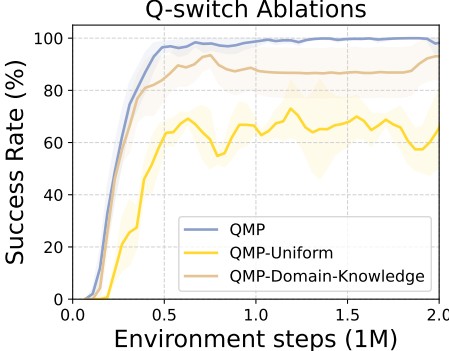

Figure 10: QMP outperforms alternate policy mixtures in Multistage Reacher.

## 8 CONCLUSION

We propose an unbiased approach to sharing behaviors via off-policy data collection in MTRL: Q-switch Mixture of Policies. We demonstrate empirically that QMP effectively improves the rate of convergence and task performance in manipulation, locomotion, and navigation tasks, and is guaranteed to be as good as the underlying RL algorithm and complementary to alternate MTRL. QMP does not assume that optimal task behaviors always coincide. Thus, its improvement magnitude is limited by the degree of shareable behaviors and the suboptimality gap that exists. At the same time, this lets QMP be unbiased and find optimal policies with convergence guarantees while being equally or more sample-efficient. Since QMP only shares behaviors via off-policy data collection, it is not applicable to on-policy RL base algorithms like PPO (Schulman et al., 2017). Promising future directions include temporally-extended behavior sharing and incorporating other forms of prior task information on shareable behaviors, such as language embeddings in instruction-following tasks.

## ACKNOWLEDGEMENTS

We thank Jesse Zhang for his assistance with writing and discussions. This work was supported by Institute of Information & communications Technology Planning & Evaluation (IITP) grant (No.RS-2019-II190075, Artificial Intelligence Graduate School Program, KAIST) and National Research Foundation of Korea (NRF) grant (NRF-2021H1D3A2A03103683, Brain Pool Research Program), funded by the Korea government (MSIT). Grace Zhang and Ayush Jain were supported partly as interns at Naver AI Lab during the initiation of the project. Shao-Hua Sun was supported by the Yushan Fellow Program by the Ministry of Education, Taiwan.

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

APPENDIX

# Table of Contents

## A  QUALITATIVE RESULTS

The qualitative result videos are provided at https://qmp-mtrl.github.io/

## B  QMP DERIVATION

Following Section 4.2, we aim to derive the mixture-switch function $f_i$ such that the mixture policy $\pi_i^{\text{mix}}$ is guaranteed to be better than the current task's policy $\pi_i$. We use the generalized policy iteration procedure (Sutton & Barto, 2018) underlying single-task SAC (Haarnoja et al., 2018): policy evaluation learns $Q$ by minimizing the bellman error on the collected data, and policy improvement follows $Q$ by minimizing the KL divergence between the new policy and the exponential of the current $Q$-function, $Q^{\pi^{\text{old}}}$, shown in Eq. 1.

In practice, the gradient updates in SAC are gradual and do not instantly achieve this optimization in Eq. 1, leaving a suboptimality gap to catch up to the Q-function. We observe that due to the potential similarity of some tasks in MTRL, this suboptimality gap can be bridged using other policies. Concretely, a mixture policy $\pi_i^{\text{mix}}$ that selects the best policy from a set of all given policy candidates, *including the current policy*, ensures that $\pi_i^{\text{mix}}$ is an improvement over $\pi_i$ for the current state $s$:

Given a set of policies $\{\pi_1 \ldots \pi_N\}$ including the current task policy $\pi_i$ and a given state $s$, consider the following mixture policy:

$$\pi_i^{\text{mix}} = \arg\min_{\pi' \in \{\pi_i, \ldots \pi_N\}} D_{\text{KL}} \left( \pi'(\cdot \mid s) \middle\| \frac{\exp(\frac{1}{\alpha} Q^{\pi_i}(s, \cdot))}{Z^{\pi_i}(s)} \right) \tag{4}$$

This $\pi_i^{\text{mix}}$ is a better policy improvement solution to Eq. 1 than $\pi_i$, because:

$$\min_{\pi' \in \{\pi_i, \ldots \pi_N\}} D_{\text{KL}} \left( \pi'(\cdot \mid s) \middle\| \frac{\exp(\frac{1}{\alpha} Q^{\pi_i}(s, \cdot))}{Z^{\pi_i}(s)} \right) \leq D_{\text{KL}} \left( \pi_i(\cdot \mid \mathbf{s}_t) \middle\| \frac{\exp(\frac{1}{\alpha} Q^{\pi_i}(\mathbf{s}_t, \cdot))}{Z^{\pi_i}(\mathbf{s}_t)} \right)$$

Now, we can simplify Eq. 4 to obtain Definition 4.2:

$$\pi_i^{\text{mix}} = \arg\min_{\pi' \in \{\pi_i, \ldots \pi_N\}} D_{\text{KL}} \left( \pi'(\cdot \mid s) \middle\| \frac{\exp(\frac{1}{\alpha} Q^{\pi_i}(s, \cdot))}{Z^{\pi_i}(s)} \right)$$

$$= \arg\min_{\pi' \in \{\pi_i, \ldots \pi_N\}} \mathbb{E}_{a \sim \pi'(\cdot \mid s)} \left[ \log \pi'(a|s) - \log \left\{ \frac{\exp(\frac{1}{\alpha} Q^{\pi_i}(s, a))}{Z^{\pi_i}(s)} \right\} \right]$$

$$= \arg\max_{\pi' \in \{\pi_i, \ldots \pi_N\}} \mathbb{E}_{a \sim \pi'(\cdot \mid s)} \left[ -\log \pi'(a|s) + \frac{1}{\alpha} Q^{\pi_i}(s, a) - \log Z^{\pi_i}(s) \right]$$

$$= \arg\max_{\pi' \in \{\pi_i, \ldots \pi_N\}} \mathbb{E}_{a \sim \pi'(\cdot \mid s)} \left[ -\log \pi'(a|s) \right] + \mathbb{E}_{a \sim \pi'(\cdot \mid s)} \left[ \frac{1}{\alpha} Q^{\pi_i}(s, a) \right]$$

$$= \arg\max_{\pi' \in \{\pi_i, \ldots \pi_N\}} \mathbb{E}_{a \sim \pi'(\cdot \mid s)} \left[ Q^{\pi_i}(s, a) \right] + \alpha \mathcal{H} \left[ \pi'(\cdot \mid s) \right]$$

Thus, the following mixture policy guarantees improvement over $\pi_i$

$$\pi_i^{\text{mix}} = \arg\max_{\pi' \in \{\pi_i, \ldots \pi_N\}} \mathbb{E}_{a \sim \pi'(\cdot \mid s)} \left[ Q^{\pi_i}(s, a) \right] + \alpha \mathcal{H} \left[ \pi'(\cdot \mid s) \right]$$

## C  QMP CONVERGENCE GUARANTEES

We derive the convergence guarantees for *mixture soft policy iteration* used in the QMP Algorithm 1. We augment the derivation of soft policy iteration in SAC (Haarnoja et al., 2018), which is our

base algorithm, with our proposed QMP's mixture policy. Soft policy iteration follows generalized policy iteration (Sutton & Barto, 2018) which refers to the general idea of repeated application of (1) policy evaluation to update the critics and (2) policy improvement based on the updated critics, until convergence. Like SAC, we consider the tabular setting and show that QMP's modification to soft policy iteration converges to the optimal policy. Further, QMP can lead to an improved policy improvement step when there are shareable behaviors between tasks, consequently improving the sample efficiency. The derivation sketch follows:

1. Soft Policy Evaluation: QMP modifies the off-policy data collection pipeline by replacing the primary task policy $\pi_i$ with the mixture policy $\pi_i^{\text{mix}}$. However, it does not affect the soft Bellman backup operator of SAC, as shown in Haarnoja et al. (2018), and therefore the $Q$ function still converges as in SAC.

2. *Mixture* Soft Policy Improvement: QMP performs policy improvement in two steps: SAC's policy update from $\pi_i^{\text{old}} \rightarrow \pi_i$ and applying the mixture of policies from $\pi_i \rightarrow \pi_i^{\text{mix}}$.

   - Soft Policy Improvement: Since QMP does *not* modify the SAC update procedure $\pi_i^{\text{old}} \rightarrow \pi_i$, we directly use SAC's guarantees of policy improvement following Lemma 2 from Haarnoja et al. (2018).

   - *Mixture* Policy Improvement: We demonstrate QMP's mixture policy $\pi_i^{\text{mix}}$ guarantees a better policy improvement over the per-task policies $\pi_i$ that compose the mixture. In Theorem C.1, we show convergence guarantee by proving that the expected return following $\pi_i^{\text{mix}}$ is better than following $\pi_i^{\text{old}}$.

3. *Mixture* Soft Policy Iteration: In Theorem C.2, we show that the repeated application of the above steps in QMP converges to an optimal policy for each task. Furthermore, the convergence rate is faster because of a greedier policy improvement due to *Mixture* Policy Improvement.

For a given stochastic policy $\pi$ and task $\mathbb{T}_i \in \{\mathbb{T}_1 \dots \mathbb{T}_N\}$, define $V_i^\pi$ as the expected return of acting with $\pi$. Given another stochastic policy $\pi'$, define $Q_i^\pi(s, \pi'(s)) = \mathbb{E}_{a \sim \pi'(s)} Q_i^\pi(s, a)$ as the expected return of acting with $\pi'$ only in **s** and thereafter with $\pi$.

**Theorem C.1** (Mixture Soft Policy Improvement). *Consider $\pi_i^{old}$ and its associated Q-function $Q_i$. Apply SAC's policy improvement $\pi_i^{old} \rightarrow \pi_i$ and then $\pi_i \rightarrow \pi_i^{mix}$ from Eq. 3. Then, $Q^{\pi_i^{mix}}(\mathbf{s}_t, \mathbf{a}_t) \geq Q^{\pi_i}(\mathbf{s}_t, \mathbf{a}_t) \geq Q^{\pi_i^{old}}(\mathbf{s}_t, \mathbf{a}_t)$ for all tasks $\mathbb{T}_i$ and for all $(\mathbf{s}_t, \mathbf{a}_t) \in \mathcal{S} \times \mathcal{A}$ with $|\mathcal{A}| < \infty$.*

*Proof.* From Soft Policy Improvement, Lemma 2 of Haarnoja et al. (2018), we have

$$\mathbb{E}_{\mathbf{a}_t \sim \pi_i} \left[ Q^{\pi_i^{\text{old}}}(\mathbf{s}_t, \mathbf{a}_t) - \log \pi_i(\mathbf{a}_t | \mathbf{s}_t) \right] \geq V^{\pi_i^{\text{old}}}(\mathbf{s}_t).$$

Rewrite the difference as $\delta(\mathbf{s}_t)$,

$$\delta(\mathbf{s}_t) = \mathbb{E}_{\mathbf{a}_t \sim \pi_i} \left[ Q^{\pi_i^{\text{old}}}(\mathbf{s}_t, \mathbf{a}_t) - \log \pi_i(\mathbf{a}_t | \mathbf{s}_t) \right] - V^{\pi_i^{\text{old}}}(\mathbf{s}_t) \geq 0.$$

From Eq. 3,

$$\pi_i^{\text{mix}} = \arg \max_{\pi' \in \{\pi_1, \dots, \pi_N\}} \mathbb{E}_{a \sim \pi'(\cdot|s)} \left[ Q^{\pi_i}(s, a) \right] + \alpha \mathcal{H} \left[ \pi'(\cdot \mid s) \right].$$

Therefore, we have a positive difference $\omega(\mathbf{s}_t)$,

$$\omega(\mathbf{s}_t) = \mathbb{E}_{\mathbf{a}_t \sim \pi_i^{\text{mix}}} \left[ Q^{\pi_i^{\text{old}}}(\mathbf{s}_t, \mathbf{a}_t) - \log \pi_i^{\text{mix}}(\mathbf{a}_t | \mathbf{s}_t) \right] - \mathbb{E}_{\mathbf{a}_t \sim \pi_i} \left[ Q^{\pi_i^{\text{old}}}(\mathbf{s}_t, \mathbf{a}_t) - \log \pi_i(\mathbf{a}_t | \mathbf{s}_t) \right] \geq 0.$$

We use $\delta$ to expand the soft Bellman equation to derive the relationship between $Q^{\pi_i^{\mathrm{old}}}$ and $Q^{\pi_i}$,

$$
\begin{aligned}
Q^{\pi_i^{\mathrm{old}}}(\mathbf{s}_t, \mathbf{a}_t) &= r(\mathbf{s}_t, \mathbf{a}_t) + \gamma \, \mathbb{E}_{\mathbf{s}_{t+1} \sim p} \left[ V^{\pi_i^{\mathrm{old}}}(\mathbf{s}_{t+1}) \right] \\
&= r(\mathbf{s}_t, \mathbf{a}_t) + \gamma \, \mathbb{E}_{\mathbf{s}_{t+1} \sim p} \left[ \mathbb{E}_{\mathbf{a}_{t+1} \sim \pi_i} \left( Q^{\pi_i^{\mathrm{old}}}(\mathbf{s}_{t+1}, \mathbf{a}_{t+1}) - \log \pi_i(\mathbf{a}_{t+1}|\mathbf{s}_{t+1}) \right) - \delta(\mathbf{s}_{t+1}) \right] \\
&\vdots \\
&= \underbrace{\sum_{k=0}^{\infty} \gamma^k \, \mathbb{E}_{\mathbf{s}_{t+k} \sim p, \, \mathbf{a}_{t+k} \sim \pi_i} \left[ r(\mathbf{s}_{t+k}, \mathbf{a}_{t+k}) - \log \pi_i(\mathbf{a}_{t+k}|\mathbf{s}_{t+k}) \right]}_{Q^{\pi_i}(\mathbf{s}_t, \mathbf{a}_t)} - \underbrace{\sum_{k=1}^{\infty} \gamma^k \, \mathbb{E}_{\mathbf{s}_{t+k} \sim p} \left[ \delta(\mathbf{s}_{t+k}) \right]}_{\Delta_1} \\
&= Q^{\pi_i}(\mathbf{s}_t, \mathbf{a}_t) - \Delta_1
\end{aligned}
$$

Likewise, we use $\delta$ and $\omega$ to derive the relationship between $Q^{\pi_i^{\mathrm{old}}}$ and $Q^{\pi_i^{\mathrm{mix}}}$,

$$
\begin{aligned}
Q^{\pi_i^{\mathrm{old}}}(\mathbf{s}_t, \mathbf{a}_t) &= r(\mathbf{s}_t, \mathbf{a}_t) + \gamma \, \mathbb{E}_{\mathbf{s}_{t+1} \sim p} \left[ V^{\pi_i^{\mathrm{old}}}(\mathbf{s}_{t+1}) \right] \\
&= r(\mathbf{s}_t, \mathbf{a}_t) + \gamma \, \mathbb{E}_{\mathbf{s}_{t+1} \sim p} \left[ \mathbb{E}_{\mathbf{a}_{t+1} \sim \pi_i} \left( Q^{\pi_i^{\mathrm{old}}}(\mathbf{s}_{t+1}, \mathbf{a}_{t+1}) - \log \pi_i(\mathbf{a}_{t+1}|\mathbf{s}_{t+1}) \right) - \delta(\mathbf{s}_{t+1}) \right] \\
&= r(\mathbf{s}_t, \mathbf{a}_t) + \gamma \, \mathbb{E}_{\mathbf{s}_{t+1} \sim p} \left[ \mathbb{E}_{\mathbf{a}_{t+1} \sim \pi_i^{\mathrm{mix}}} \left( Q^{\pi_i^{\mathrm{old}}}(\mathbf{s}_{t+1}, \mathbf{a}_{t+1}) - \log \pi_i^{\mathrm{mix}}(\mathbf{a}_{t+1}|\mathbf{s}_{t+1}) \right) - \delta(\mathbf{s}_{t+1}) - \omega(\mathbf{s}_{t+1}) \right] \\
&\vdots \\
&= \underbrace{\sum_{k=0}^{\infty} \gamma^k \, \mathbb{E}_{\mathbf{s}_{t+k} \sim p, \, \mathbf{a}_{t+k} \sim \pi_i^{\mathrm{mix}}} \left[ r(\mathbf{s}_{t+k}, \mathbf{a}_{t+k}) - \log \pi_i^{\mathrm{mix}}(\mathbf{a}_{t+k}|\mathbf{s}_{t+k}) \right]}_{Q^{\pi_i^{\mathrm{mix}}}(\mathbf{s}_t, \mathbf{a}_t)} \\
&\quad - \underbrace{\sum_{k=1}^{\infty} \gamma^k \, \mathbb{E}_{\mathbf{s}_{t+k} \sim p} \left[ \delta(\mathbf{s}_{t+k}) \right]}_{\Delta_2} - \underbrace{\sum_{k=1}^{\infty} \gamma^k \, \mathbb{E}_{\mathbf{s}_{t+k} \sim p} \left[ \omega(\mathbf{s}_{t+k}) \right]}_{\Omega} \\
&= Q^{\pi_i^{\mathrm{mix}}}(\mathbf{s}_t, \mathbf{a}_t) - \Delta_2 - \Omega,
\end{aligned}
$$

We assume that the effect of the difference $\Delta_2 - \Delta_1$ due to different state coverage is lower compared to the effect of $\Omega$ because $\omega$ is accumulated at every state, i.e., $\Delta_2 + \Omega = \Delta_1 + (\Delta_2 - \Delta_1) + \Omega \geq \Delta_1$

Since $\Delta_1, \Delta_2 \geq 0$ and $\Omega \geq 0$, we have

$$
Q^{\pi_i^{\mathrm{mix}}}(\mathbf{s}_t, \mathbf{a}_t) \geq Q^{\pi_i}(\mathbf{s}_t, \mathbf{a}_t) \geq Q^{\pi_i^{\mathrm{old}}}(\mathbf{s}_t, \mathbf{a}_t)
$$

$\square$

**Theorem C.2** (Mixture Soft Policy Iteration). *Repeated application of (i) soft policy evaluation and (ii) mixture soft policy improvement (Theorem C.1) to any $\pi_i \in \Pi$ converges to an optimal policy $\pi_i^*$ such that $Q_i^{\pi_i^*}(\mathbf{s}_t, \mathbf{a}_t) \geq Q_i^{\pi_i}(\mathbf{s}_t, \mathbf{a}_t)$ for all $\pi_i \in \Pi$ and $(\mathbf{s}_t, \mathbf{a}_t) \in \mathcal{S} \times \mathcal{A}$ with $|\mathcal{A}| < \infty$. Furthermore, the sample efficiency and rate of convergence is at least as good as SAC in the presence of mixture policy improvement.*

*Proof.* Let $\pi_i^k$ be the policy at iteration $k$. By SAC's soft policy iteration, the sequence $Q_i^{\pi_i^k}$ is monotonically increasing, because $\pi_i^{\mathrm{mix}}$ only modifies the online data collected and SAC is an off-policy algorithm. Thus, Theorem 1 (Soft Policy Iteration) from Haarnoja et al. (2018) Appendix B.3 directly applies here and proves that repeated application of soft policy evaluation and soft policy improvement converges to an optimal policy $\pi_i^*$.

Mixture soft policy improvement (Theorem C.1) shows that $\pi_i^{\mathrm{mix}}$ is a greedier policy improvement over $\pi_i$ with respect to each estimate of $Q_i^{\pi_i^k}$. Thus, the expected returns in the data collected by QMP policy, $Q_i^{\pi_i^{\mathrm{mix};k}}$, is greater than or equal to that collected by the individual task policy, $Q_i^{\pi_i^k}$. Therefore,

every mixture soft policy improvement step constitutes a *larger policy improvement step* than SAC's soft policy improvement step. This makes the convergence of mixture soft policy iteration (repeated application of soft policy evaluation and Theorem C.1) an improvement over soft policy iteration.

□

# D ENVIRONMENT DETAILS

## D.1 MULTISTAGE REACHER

We implement multistage reacher tasks on the Open AI Gym (Brockman et al., 2016) Reacher environment simulated in the MuJoCo physics engine (Todorov et al., 2012) by defining a sequence of 3 subgoals per task, as specified in Table 1. For all tasks, the reacher is initialized at the same start position with a small random perturbation sampled uniformly from $[-0.01, 0.01]$ for each coordinate. The observation includes the agent's proprioceptive state and how many subgoals have been reached but not subgoal locations, as they must be inferred from the respective task's reward function.

We set up the tasks to ensure that we can evaluate behavior sharing when the task rewards are qualitatively different (see Figure 6a):

- For every task except Task 3, the reward function is the default gym reward function based on the distance to the goal, plus an additional bonus for every subgoal completed.
- For Task 1, the reward is shifted by -2 at every timestep.
- Task 3 receives only a sparse reward of 1 for every subgoal reached.
- Task 4 has one fixed goal set at its initial position.

|  | Subgoal 1 | Subgoal 2 | Subgoal 3 |
|---|---|---|---|
| $T_0$ | (0.2, 0.3, 0.5) | (0.3, 0, 0.3) | (0.4, -0.3, 0.4) |
| $T_1$ | (0.2, 0.3, 0.5) | (0.3, 0, 0.3) | (0.4, 0.3, 0.2) |
| $T_2$ | (0.3, 0, 0.3) | (0.4, 0.3, 0.2) | (0.4, -0.3, 0.4) |
| $T_3$ | (0.3, 0, 0.3) | (0.4, -0.3, 0.4) | (0.2, 0.3, 0.5) |
| $T_4$ | initial | initial | initial |

Table 1: Coordinates of subgoal locations for each task in Multistage Reacher. Shared subgoals are highlighted in the same color. For Task 4, the only goal is to stay in the initial position.

**QMP-Domain**: Section 7.4 ablates the importance of an adaptive and state-dependent Q-switch by replacing it with a domain-dependent distribution over other tasks based on apparent task similarity. Specifically, to define the mixture probabilities for QMP-Domain in Multistage Reacher, we use the domain knowledge of the subgoal locations of the tasks to determine the mixture probabilities. We use the ratio of *shared sub-goal sequences* between each pair of tasks (not necessarily the shared subgoals) over the total number of sub-goal sequences, 3, to calculate how much behavior must be shared between two tasks. For that ratio of shared behavior, we distribute the probability mass uniformly between all task policies that share that behavior. For Task 4, the conflicting task, we do not do any behavior sharing and only use $\pi_4$ to gather data.

Each Task $\mathbb{T}_i$ consists of 3 sub-goal sequences $\{S_0, S_1, S_2\}$ (e.g. [initial → Subgoal 1], [Subgoal 1 → Subgoal 2], and [Subgoal 2 → Subgoal 3]). For each sequence $s \in \{S_0, S_1, S_2\}$, we divide equally the contribution of each task $\mathbb{T}_j$'s policy $\pi_j$ that shares the sequence $s$ (i.e. if $\mathbb{T}_0$ and $\mathbb{T}_1$ both contain sequence $s$, where we use the notation $\mathbb{1}(s \in \mathbb{T}_i)$ as the indicator function for whether Task $\mathbb{T}_i$ contains sequence $s$, then $\pi_0$ and $\pi_1$ both have $\frac{1}{2}$ contribution for $s$). Each sequence contributes equally to the overall mixture probabilities for Task $i$ (i.e. all policies that shares sequence $S_i$ contributes in total $\frac{1}{3}$ to the mixture probability for the behavior policy of Task $\mathbb{T}_i$). Thus, the contribution probability of Policy $\pi_j$ to Task $\mathbb{T}_i$ is:

$$p_{j \to i} = \sum_{s \in \{S_0, S_1, S_2\}} \frac{1}{3} \cdot \frac{\mathbb{1}(s \in \mathbb{T}_j)}{\sum_k \mathbb{1}(s \in \mathbb{T}_k)}$$

$$\pi_i^{\text{mix}} = \sum_j p_{j \to i} \, \pi_j$$

Reusing notation for mixture probabilities, we have,

$$\pi_0^{mix} = \frac{2}{3}\pi_0 + \frac{1}{3}\pi_1$$
$$\pi_1^{mix} = \frac{1}{3}\pi_0 + \frac{2}{3}\pi_1$$
$$\pi_2^{mix} = \frac{5}{6}\pi_2 + \frac{1}{6}\pi_3$$
$$\pi_3^{mix} = \frac{1}{6}\pi_2 + \frac{5}{6}\pi_3$$
$$\pi_4^{mix} = \pi_4$$

### D.2 MAZE NAVIGATION

The layout and dynamics of the maze follow Fu et al. (2020), but since their original design aims to train a single agent to reach a fixed goal from multiple start locations, we modified it to have both start and goal locations fixed in each task, as in Nam et al. (2022). The start location is still perturbed with a small noise to avoid memorizing the task. The observation consists of the agent's current position and velocity. But, it lacks the goal location, which should be inferred from the dense reward based on the distance to the goal. The action space is the target 2D velocity of the point mass agent.

The layout we used is LARGE_MAZE which is an $8\times11$ maze with paths blocked by walls. The complete set of 10 tasks is visualized in Figure 12, where green and red spots correspond to the start and goal locations, respectively. The environment provides an agent a dense reward of $\exp(-dist)$ where $dist$ is a linear distance between the agent's current position and the goal location. It also gives a penalty of 1 at each timestep in order to prevent the agent from exploiting the reward by staying near the goal. The episode terminates either as soon as the goal is reached by having $dist < 0.5$ or when 600 timesteps have passed.

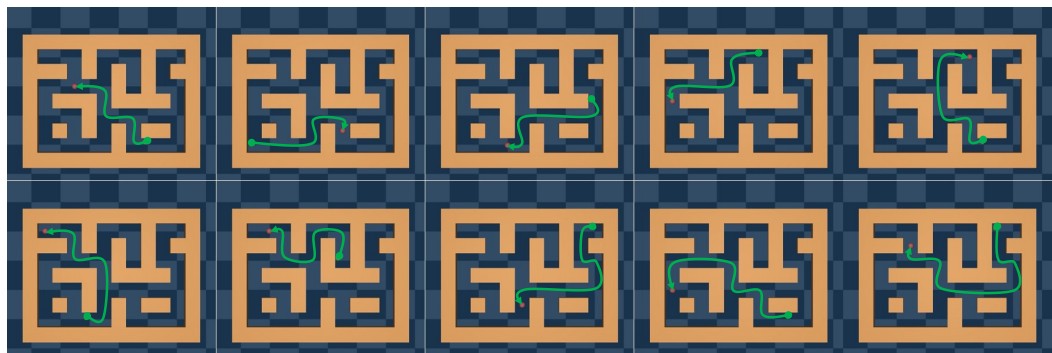

Figure 12: Ten tasks defined for the Maze Navigation. The start and goal locations in each task are shown in green and red spots, respectively, and an example path is shown in green.

### D.3 META-WORLD MANIPULATION

For Meta-World CDS, we reproduce the Meta-world environment proposed by Yu et al. (2021) using the Meta-world codebase (Yu et al., 2019), where the door and drawer are both placed side-by-side on the tabletop for all tasks (see Figure 6c). The observation space consists of the robot's proprioceptive state, the drawer handle state, the door handle state, and the goal location, which varies based on the task. Unlike Yu et al. (2021), we additionally remove the previous state from the observation space so the policies cannot easily infer the current task, making it a challenging multi-task environment. The environment also uses the default Meta-World reward functions which is composed of two distance-based rewards: distance between the agent end effector and the object, and distance between the object and its goal location. We use this modified environment instead of the Meta-world benchmark because our problem formulation of simultaneous multi-task RL requires

a consistent environment across tasks. For Meta-World MT10, we directly use the implementation provided in (Yu et al., 2019) without changes.

In both cases, the observation space consists of the robot's proprioceptive state, locations for objects present in the environment (ie. door and drawer handle for CDS, the single target object location for MT10) and the goal location. In Meta-World CDS, due to the shared environment, there are no directly conflicting task behaviors, since the policies either go to the door or the drawer, they should ignore the irrelevant behaviors of policies interacting with the other object. In Meta-World MT10, each task interacts with a different object but in an overlapping state space so there is a mix of shared and conflicting behaviors.

### D.4  WALKER2D

Walker2D is a 9 DoF bipedal walker agent with the multi-task set of 4 tasks proposed and implemented by Lee et al. (2019): walking forward at a target velocity, walking backward at a target velocity, balancing under random external forces, and crawling under a ceiling. Each of these tasks involves different gaits or body positions to accomplish successfully without any obviously identifiable shared behavior in the optimal policies. Behavior sharing can still be effective during training to aid exploration and share helpful intermediate behaviors, like balancing. However, there is no obviously identifiable conflicting behavior either in this task set. Because each task requires a different gait, it is unlikely for states to recur between tasks and even less likely for states that are shared to require conflicting behaviors. For instance, it is common for all policies to struggle and fall at the beginning of training, but all tasks would require similar stabilizing and correcting behavior over these states.

### D.5  KITCHEN

We modify the Franka Kitchen environment proposed by Gupta et al. (2019) and based on the implementation from Fu et al. (2020). Since this environment is typically used for long horizon or offline RL, we chose shorter tasks that are learnable with online RL. Furthermore, we added a dense reward based on the Meta-World reward function. We formed our 10 task MTRL set by choosing 10 available tasks in the kitchen environment that interacted with the same objects: turning the top burner on or off, turning the bottom burner on or off, turning the light switch on and off, open or closing the sliding cabinet, and opening and closing the hinge cabinet. The observation space consists of the robot's state, the location of the target object, and the goal location for that object. Similar to the Meta-World CDS environment, these tasks may share behaviors navigating around the kitchen to the target object but have plenty of irrelevant behavior between tasks that interact with different objects and conflicting behaviors when opening or closing the same object.

## E  ADDITIONAL RESULTS

### E.1  MULTISTAGE REACHER PER TASK RESULTS

Additional results and analysis on Multistage Reacher are shown in Figure 13. QMP outperforms all the baselines in this task set, as shown in Figure 8. Task 3 receives only a sparse reward and, thus, can benefit the most from shared exploration. We observe that QMP gains the most performance boost due to selective behavior-sharing in Task 3. The No-Shared-Behavior baseline is unable to solve Task 3 at all due to its sparse reward nature. The other baselines which share uniformly suffer at Task 3, likely because they also share behaviors from other conflicting tasks, especially Task 4. We explore this further in the following Section F.

For all tasks, QMP outperforms or is comparable to No-Shared-Behavior, which shows that selective behavior-sharing can help accelerate learning when task behaviors are shareable and is robust when tasks conflict. Fully-Shared-Behavior especially underperforms in Tasks 2 and 3, which require conflicting behaviors upon reaching Subgoal 1, as defined in Table 1. In contrast, it excels at the beginning of Task 0 and Task 1 as their required behaviors are completely shared. However, it suffers after Subgoal 2, as the task objectives diverge.

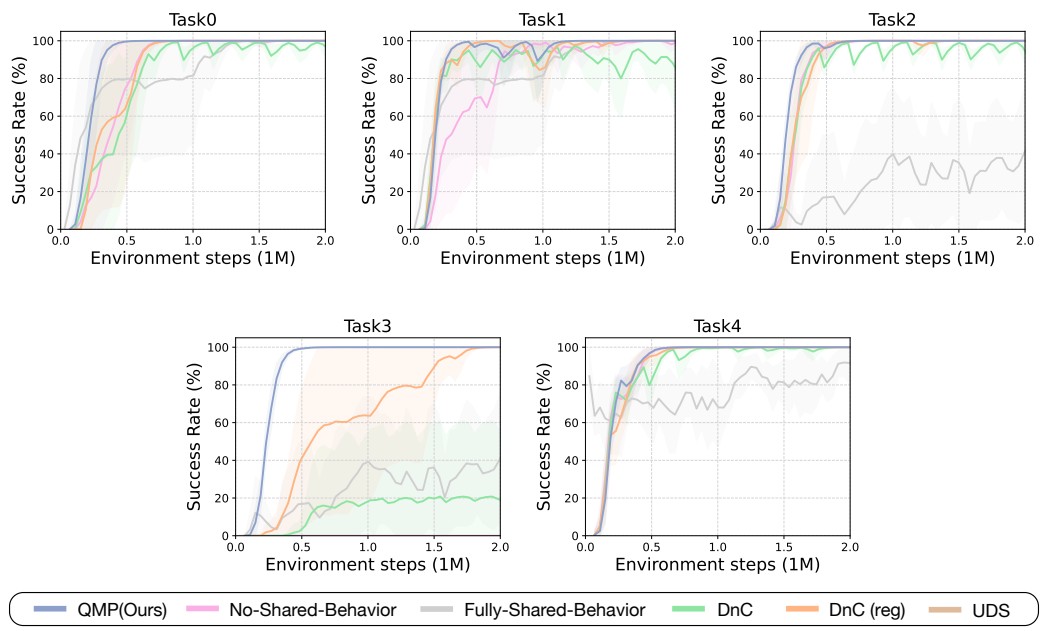

Figure 13: Success rates for individual tasks in Multistage Reacher. Our method especially helps in learning Task 3, which requires extra exploration because it only receives a sparse reward.

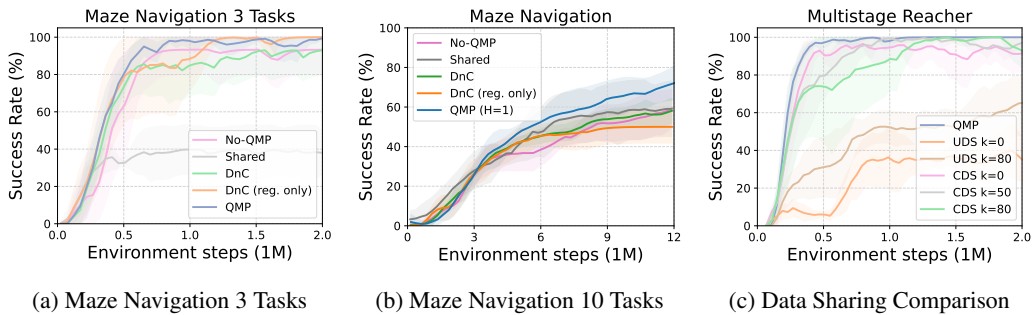

(a) Maze Navigation 3 Tasks      (b) Maze Navigation 10 Tasks      (c) Data Sharing Comparison

Figure 14: QMP scales well from (a) 3 tasks to (b) 10 tasks in Maze Navigation, especially in comparison to other behavior sharing methods. (c) Online data sharing is very efficient when given task reward functions (all CDS versions), but suffers without (all UDS versions).

### E.2 QMP SCALES WITH TASK SET SIZE IN MAZE NAVIGATION

We look at the behavior sharing methods in the Maze Navigation task for a task set with 3 tasks (Figure 14a) and 10 tasks (Figure 14b) and see that QMP scales well from 3 to 10 tasks, even compared to other behavior sharing methods. Similar to Meta-World, we hypothesize QMP scales better with a larger task set size of similar tasks due to there being more shareable behaviors between tasks. We see that by selectively sharing behaviors, QMP is able to identify and share helpful behaviors in the larger tasks sets whereas other behavior sharing methods struggle.

### E.3 QMP OUTPERFORMS DATA SHARING WITH REWARD LABELING

In Figure 14c, we report multiple sharing percentiles for UDS and for CDS (Yu et al., 2021) which assumes access to ground truth task reward functions which it uses to re-label the shared data. When the shared data is relabeled with task reward functions, thereby bypassing the conflicting behavior problem, online data sharing approaches can work very well. But when unsupervised, we see that online data sharing can actually harm performance in environments with conflicting tasks, with

the more conservative data sharing approach (UDS k=80) out-performing sharing all data. $k$ is the percentile above with we share a transition between tasks, with higher $k$ representing more conservative data sharing. Full details on our online UDS and CDS implementation are in Section H.6

### E.4 PCGRAD RESULTS

We evaluate whether QMP combined with PCGrad (Yu et al., 2020) results in complementary benefits. PCGrad is a popular MTRL algorithm that learns a policy with shared parameters and alleviates negative interference between tasks by modifying the multi-task gradients. In Figure 15 and Table 2 , we see that QMP + PCGrad significantly improves PCGrad performance in 3 out of 4 environments.

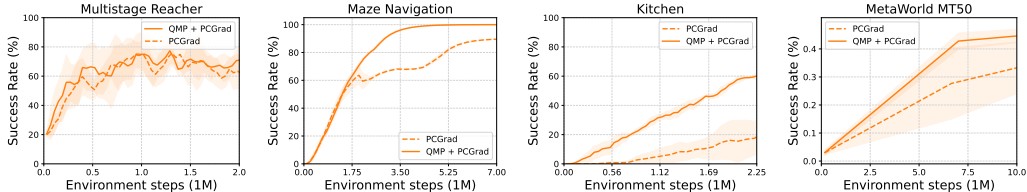

Figure 15: Combining QMP with PCGrad yields complementary improvement in 3 out of the 4 environments we tested on. Dashed lines are PCGrad only and solid lines are QMP + PCGrad.

| Approach | Reacher | Maze | Kitchen | Meta-World 50 |
|---|---|---|---|---|
| PCGrad | **0.78** | 0.90 | 0.55 | 0.35 |
| QMP + PCGrad | **0.78** | **1.00** | **0.60** | **0.42** |

Table 2: QMP improves performance of PCGrad across various benchmarks

### E.5 ADDITIONAL MTRL COMPARISONS

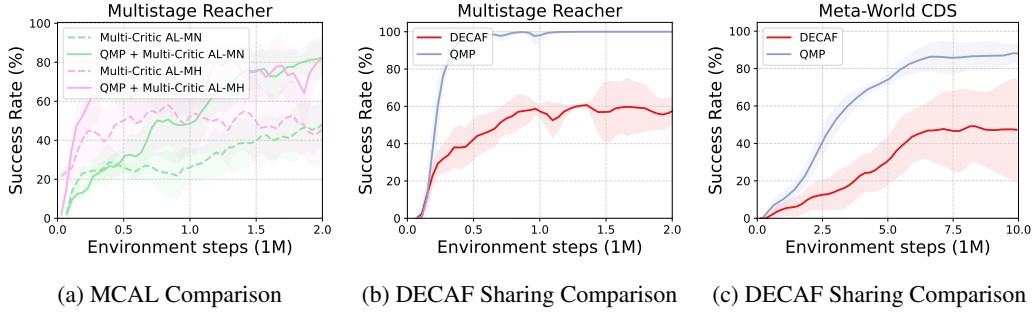

(a) MCAL Comparison          (b) DECAF Sharing Comparison          (c) DECAF Sharing Comparison

Figure 16: Combining our method with another parameter sharing method, MCAL, shows complementary benefits in (a). Our method outperforms DECAF in Multistage Reacher (b) and Meta-World CDS(c), demonstrating that learning to directly use Q-functions from other tasks is more challenging and sample inefficient than using the current task's Q-function to evaluate other tasks' policies.

Multi-Critic Actor Learning (MCAL) (Mysore et al., 2022) is a parameter sharing MTRL method that aims to tackle potential negative interference between tasks by learning separate critics for each task while training a single multi-task actor. We add QMP to two variants of MCAL, Multi-Critic AL-MN which maintains separate networks for each critic and Multi-Critic AL-MH which uses a single multi-head network for the critic, in Multistage Reacher in Figure 16a. We see that adding QMP provides around a 20% final success rate gain in both variants and is more sample efficient.

We also compare our method with DECAF (Glatt et al., 2020), a MTRL method which shared Q-functions between tasks instead of behavioral policies. DECAF learns task specific weights to linearly combine the task Q-functions which is used to train the task policy. In contrast, our method uses the task Q-function to evaluate different tasks' policies to incorporate into the task's behavioral

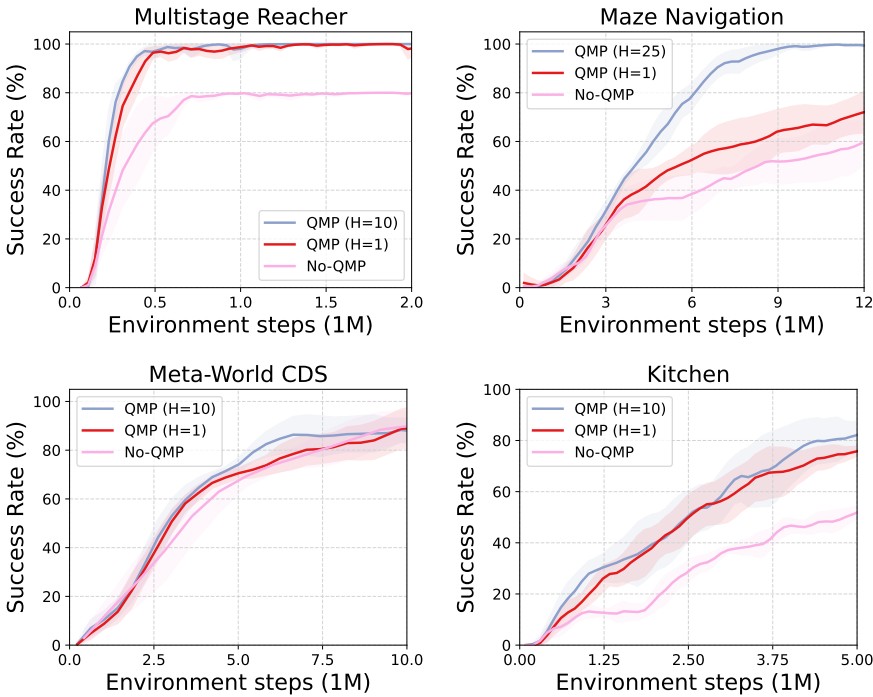

Figure 17: In each case above, QMP with H-step rollouts of the behavioral policy (blue) performs no worse than QMP with 1-step rollouts (red), with the H-step rollouts helping significantly in some tasks. Additionally both versions of QMP outperform the No-QMP baseline.

policy. Our method only modifies the data collection process, not the RL objective, and does not have a learned component. In Multistage Reacher (Figure 16b) and Meta-World CDS (Figure 16c), we see that QMP outperforms DECAF by more that 20% final success rate.

### E.6 TEMPORALLY-EXTENDED BEHAVIOR SHARING

Motivated by prior work in heirarchical RL (Machado et al., 2017; Jinnai et al., 2019b;a; Hansen et al., 2019; Zhang et al., 2020) and skill learning (Pertsch et al., 2021) , we explore temporally extended behavior sharing by simply following the actions of the policy $\pi_j$ selected by $\pi^i_{mix}$ for $H$ steps before re-evaluating $\pi^i_{mix}$. Furthermore, a recent work Xu et al. (2024) provides theoretical results that shows myopic ($\epsilon$-greedy) policy sharing can be sample efficient in sufficiently diverse multi-task settings, providing theoretical support for temporally extended multi-task behavior sharing in some settings.

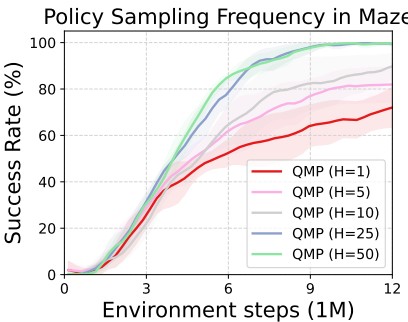

Figure 18: QMP consistently improves performance as $H$ increases in Maze.

We study the effect of sharing temporally extended behaviors of length $H$ in Maze Navigation in Figure 18, by rolling out the chosen task policy for 1, 5, 10, 25, and 50 timesteps. We see that performance improves when sharing longer behaviors (25 and 50 timesteps) which are more coherent and temporally extended. This is true even though we choose the behavioral policy greedily, only evaluating the current state $s$ every $H$ steps. Importantly, the guarantees from Theorem C.1 do not extend to $H$-step policy roll-outs and increasing $H$ does not help in all environments. We compare the performance of No-QMP, QMP, and QMP with temporally extended behavior sharing where we choose the best performance out of $H = 10$ and $H = 25$ in Table 3 and Figure 17. Nevertheless, the impressive results in Maze suggest that multi-task temporally extended behavior sharing is worth exploring in future work.

Table 3: Temporally Extended Behavior Sharing

| Environment | H-value | No-QMP | QMP | QMP (H>1) |
|---|---|---|---|---|
| Reacher | 10 | 80 ± 0 | **100** ± 0 | **100** ± 0 |
| Maze | 25 | 57.9 ± 0.09 | 72.9 ± 0.1 | **99.9** ± 0.0 |
| MT-CDS | 10 | 97.5 ± 4.5 | 93.7 ± 8.5 | **98.8** ± 2.0 |
| MT10 | 10 | 79.1 ± 5.97 | **89.0** ± 0.01 | 82. ± 4.48 |
| Kitchen | 10 | 65.5 ± 11.0 | 77.3 ± 5.3 | **84.5** ± 8.7 |
| Walker | 10 | 3110 ± 220 | 3205 ± 218 | **3310** ± 203 |

# F  QMP BEHAVIOR SHARING ANALYSIS

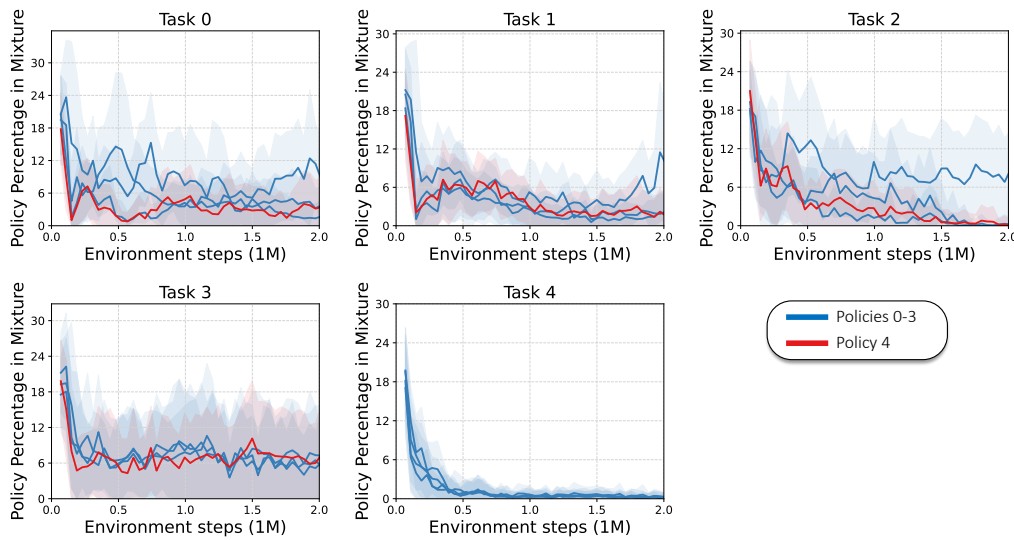

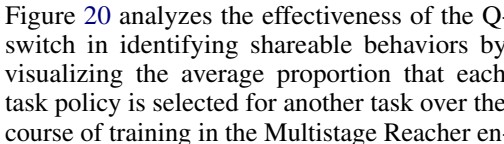

Figure 19: Mixture probabilities per task of other policies over the course of training for Multistage Reacher. The conflicting task Policy 4, which requires staying stationary, is highlighted in red.

**QMP learns to not share from conflicting tasks**: We visualize the mixture probabilities per task of other policies in Figure 19 for Multistage Reacher, highlighting the conflicting Task 4 in red. Throughout training, we see that QMP learns to share less behavior from Policy 4 than other policies in Tasks 0-3 and shares the least total cross-task behavior in Task 4. This supports our claim that the Q-switch can identify conflicting behaviors that should not be shared. We note that Task 3 has a relatively larger amount of sharing than other tasks. Since Task 3 has sparse rewards, it benefits the most from exploration via selective behavior-sharing from other tasks.

Figure 20 analyzes the effectiveness of the Q-switch in identifying shareable behaviors by visualizing the average proportion that each task policy is selected for another task over the course of training in the Multistage Reacher en-

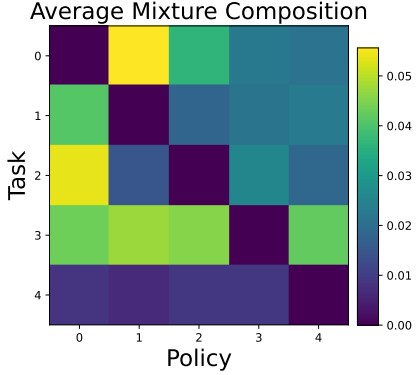

Figure 20: Average contribution of each Policy $j$ (col $j$) in each Task $i$'s (row $i$) data collection on Reacher Multistage (diagonal zeroed for contrast).

vironment. This average mixture composition statistic intuitively analyzes whether QMP identifies shareable behaviors between similar tasks and avoids behavior sharing between conflicting or irrelevant tasks. As we expect, the Q-switch for Task 4 utilizes the least behavior from other policies

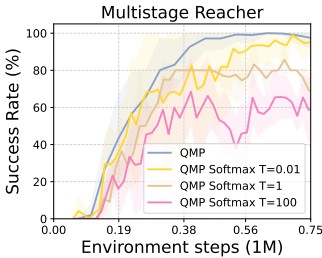 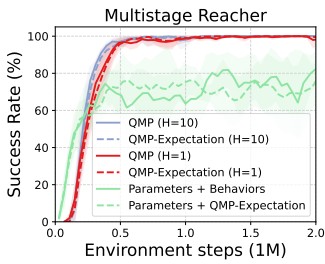 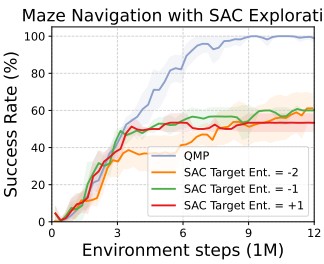

(a) Probabilistic Mixture Ablation  (b) Expected Q-value Approxima-    (c) Single-Task Exploration
                                           tions

Figure 21: (a) Using probabilistic mixtures with QMP by using a softmax over Q values with temperature T, which determines the spread of the distribution. (b)Across different QMP versions, evaluating mean policy actions (solid lines) vs. sampling 10 actions to estimate expected Q-values (dashed lines) result in similar performance. (c) Single-task exploration by varying SAC target entropy. QMP reaches a higher success rate because it shares exploratory behavior **across** tasks.

(bottom row), and Policy 4 shares the least with other tasks (rightmost column). Since the agent at Task 4 is rewarded to stay at its initial position, this behavior conflicts with all the other goal-reaching tasks. Of the remaining tasks, Task 0 and 1 share the most similar goal sequence, so it is intuitive why they benefit from shared exploration and are often selected by their respective Q-switches. Finally, unlike the other tasks, Task 3 receives only a sparse reward and therefore relies heavily on shared exploration. In fact, QMP demonstrates the greatest advantage in this task (Appendix Figure 13).

**Behavior-sharing reduces over training**: Figure 19 shows that the total amount of behavior-sharing decreases over the course of training in all tasks, which demonstrates a naturally arising preference in the Q-switch for the task-specific policy as it becomes more proficient at its own task.

### F.1  QUALITATIVE VISUALIZATION OF BEHAVIOR-SHARING

We qualitatively analyze behavior sharing by visualizing a rollout of QMP during training for the Drawer Open task in Meta-World Manipulation (Figure 9b). To generate this visualization, we use a QMP rollout during training before the policy converges to see how behaviors are shared and aid learning. For clarity, we first subsample the episodes timesteps by 10 and only report timesteps when the activated policy changes to a new one (ie. from timestep 80 to 110, QMP chose the Drawer Open policy). We qualitatively break down the episode into when the agent is approaching the drawer (top row; Steps 1-60), grasping the handle (top row; Steps 61-80), and pulling the drawer open (bottom row). This allows us to see that it switches between all task policies as it approaches the drawer, uses drawer-specific policies as it grasps the handle, and opening-specific policies as it pulls the drawer open. This suggests that in addition to ignoring conflicting behaviors, QMP is able to identify helpful behaviors to share. We note that QMP is not perfect at policy selection throughout the entire rollout, and it is also hard to interpret these shared behaviors exactly because the policies themselves are only partially trained, as this rollout is from the middle of training. However, in conjunction with the overall results and analysis, this supports our claim that QMP can effectively identify shareable behaviors between tasks.

## G  ADDITIONAL ABLATIONS AND ANALYSIS

### G.1  PROBABILISTIC MIXTURE V/S ARG-MAX

A probabilistic mixture of policies is a design choice of our approach where arg-max is replaced with softmax. However, in our initial experiments, we found no significant improvement in performance and it came with an additional hyperparameter of tuning the temperature coefficient. As we see in Figure 21a, QMP actually outperforms a probabilistic mixture over a range of softmax temperatures, justifying the design choice of argmax over softmax due to its reliable performance and simplicity.

### G.2 APPROXIMATION EXPECTED Q-VALUE OVER POLICY ACTION DISTRIBUTION

QMP's behavior policy is defined as $\pi_i^{\text{mix}} = \underset{\pi_j \in \{\pi_1,...,\pi_N\}}{\arg\max} \mathbb{E}_{a \sim \pi_j(s)} Q_i(s, a)$, which picks the task policy with the best expected Q value over its action distribution. We approximate the expectation by evaluating the Q-value of only the mean of each policy's action distribution which is computationally cheaper $\pi_i^{\text{mix}} \approx \underset{\pi_j \in \{\pi_1,...,\pi_N\}}{\arg\max} Q_i(s, \mathbb{E}_{a \sim \pi_j(s)}[a])$. We compare this to a empirical estimate that samples 10 actions from the policy distribution and picks the policy with highest average Q-value in Figure 21b, and find no significant performance difference between the two approximations. This validates that our simple approximation works well in practice, which we hypothesize is due to the low variance of the task policies.

### G.3 QMP V/S INCREASING SINGLE TASK EXPLORATION

Since QMP seeks to gather more informative training data for the task by modifying the behavioral policy, it can be viewed as a form of multi-task exploration. We briefly investigate how single task exploration differs from multi-task exploration by tuning the target entropy in SAC in Figure 21c which influences the policy entropy and therefore exploration. We see that while tuning this exploration parameter affects the sample efficiency by more quickly learning each individual task, QMP achieves a higher final success rate by incorporating behaviors form other tasks, and therefore doing multi-task exploration. The benefit of exploration or behavior sharing algorithms specialized for multi-task RL is precisely this ability to transfer and share information between tasks.

### G.4 QMP RUNTIME

While QMP does require more policy and q-function evaluations to sample from $\pi_{mix}^i$ in comparison to the base RL method, these evaluations can be greatly parallelized and do not significantly increase runtime (see Table 4) for average runtimes for our experiments). Each sample from $\pi_{mix}^i$ requires querying $N$ policy proposals and $N$ Q-values. In QMP + Parameter-Sharing, thanks to the multihead architectures of the policy and Q-networks, all N evaluations are done in one single pass. Thus, with two passes through neural networks, we can get N action candidates and their N Q-values. Therefore, the increase in time is negligible. Even without parameter-sharing, $Q_i(s, a_j)$ evaluations can be batched $\forall j$ and the policy evaluations $\pi_j(a_j|s)$ are all independent, and can be obtained in parallel. In our implementation, we batch the Q evaluations, but do not parallelize the policy evaluations.

Table 4: Runtime Comparison

| Environment | No-Sharing | QMP + No-Sharing | Parameter-Sharing | QMP + Parameter-Sharing |
|---|---|---|---|---|
| Reacher Multistage | 12.5 hr | 14.2 hr | 14 hr | 16.2 hr |
| MT50 | – | – | 7 days, 3hr | 7 days, 6 hr |

## H IMPLEMENTATION DETAILS

The SAC implementation we used in all our experiments is based on the open-source implementation from Garage (garage contributors, 2019). We used fully connected layers for the policies and Q-functions with the default hyperparameters listed in Table 5. For DnC baselines, we reproduced the method in Garage to the best of our ability with minimal modifications.

We used PyTorch (Paszke et al., 2019) for our implementation. We run the experiments primarily on machines with either NVIDIA GeForce RTX 2080 Ti or RTX 3090. Most experiments take around one day or less on an RTX 3090 to run. We use the Weights & Biases tool (Biewald, 2020) for logging and tracking experiments. All the environments were developed using the OpenAI Gym interface (Brockman et al., 2016).

## H.1 Hyperparameters

Table 5 details the list of important hyperparameters on all the 3 environments. For all environments, we used a 2 layer fully connected network with hidden dimension 256 and a tanh activation function for the policies and Q functions. We use a target network for the Q function with target update $\tau = 0.995$ and trained with an RL discount of $\gamma = 0.99$.

Table 5: Hyperparameters.

| Hyperparameter | Multistage Reacher | Maze Navigation | Meta-World CDS |
|---|---|---|---|
| Minimum buffer size (per task) | 10000 | 3000 | 10000 |
| # Environment steps per update (per task) | 1000 | 600 | 500 |
| # Gradient steps per update (per task) | 100 | 100 | 50 |
| Batch size | 32 | 256 | 256 |
| Learning rates for $\pi$, $Q$ and $\alpha$ | 0.0003 | 0.0003 | 0.0015 |

| Hyperparameter | Meta-World MT10 | Walker | Kitchen |
|---|---|---|---|
| Minimum buffer size (per task) | 500 | 2500 | 200 |
| # Environment steps per update (per task) | 500 | 1000 | 200 |
| # Gradient steps per update (per task) | 50 | 1500 | 50 |
| Batch size | 2560 | 256 | 1280 |
| Learning rates for $\pi$, $Q$ and $\alpha$ | 0.0015 | 0.0003 | 0.0003 |

## H.2 No-Shared-Behaviors

All $N$ networks have the same architecture with the hyperparameters presented in Table 5.

## H.3 Fully-Shared-Behaviors

Since it is the only model with a single policy, we increased the number of parameters in the network to match others and tuned the learning rate. The hidden dimension of each layer is 600 in Multistage Reacher, 834 in Maze Navigation, and 512 in Meta-World Manipulation, and we kept the number of layers at 2. The number of environment steps as well as the number of gradient steps per update were increased by $N$ times so that the total number of steps could match those in other models. For the learning rate, we tried 4 different values (0.0003, 0.0005, 0.001, 0.0015) and chose the most performant one. The actual learning rate used for each experiment is 0.0003 in Multistage Reacher and Maze Navigation, and 0.001 in Meta-World Manipulation.

This modification also applies to the Shared Multihead baseline, but with separate tuning for the network size and learning rates. In Multistage Reacher, we used layers with hidden dimensions of 512 and 0.001 as the final learning rate. In Maze Navigation, we used 834 for hidden dimensions and 0.0003 for the learning rate.

## H.4 DnC

We used the same hyperparameters as in Separated, while the policy distillation parameters and the regularization coefficients were manually tuned. Following the settings in the original DnC (Ghosh et al., 2018), we adjusted the period of policy distillation to have 10 distillations over the course of training. The number of distillation epochs was set to 500 to ensure that the distillation is completed. The regularization coefficients were searched among 5 values (0.0001, 0.001, 0.01, 0.1, 1), and we chose the best one. Note that this search was done separately for DnC and DnC with regularization only. For DnC, the coefficients we used are: 0.001 in Multistage Reacher and Maze Navigation, and 0.001 in Meta-World Manipulation. For Dnc with regularization only, the values are: 0.001 in Multistage Reacher, 0.0001 in Maze Navigation, and 0.001 in Meta-World Manipulation.

## H.5 QMP (OURS)

Our method also uses the default hyperparameters. QMP does not require any task specific hyperparameters. The exception is Meta-World MT10, where we found it helpful to have more conservative behavior sharing by choosing the task-specific policy 70% of the time. The remaining 30% we use the Q-filter to select a policy as usual.

Like in Baseline Multihead (Parameters-Only), the QMP Multihead architecture (Parameters+Behaviors) also required a separate tuning. Since QMP Multihead effectively has one network, we increased the network size in accordance with Baseline Multihead and tuned the learning rate in addition to the mixture warmup period. The best-performing combinations of these parameters we found are 0 and 0.001 in Multistage Reacher, and 100 and 0.0003 in Maze Navigation, respectively.

## H.6 ONLINE UDS

Yu et al. (2022) proposes an offline multi-task RL method (UDS) that shares data between tasks if their conservative Q value falls above the $k^{th}$ percentile of the task data. Specifically, before training, you would go through all the tasks' data and share some data from Task $j$ to Task $i$ if the Task $i$ Q value of that data is greater than $k\%$ of the Q values of Task $i$'s data. UDS does not require access to task reward functions like other data-sharing approaches. It simply re-labels any shared data with the minimum task reward, making it applicable to our problem setting as we also do not assume that reward relabeling is possible.

In order to adapt UDS to online RL, instead of doing data sharing once on the given multi-task dataset, we apply UDS data sharing before every training iteration to the data in the multi-task replay buffers. Concretely, we implement this on-the-fly for every batch of sampled data by sampling one batch of data from Task $i$'s replay buffer, $\beta_i$, and one batch of data from the other task's replay buffers $\beta_{j \neq i}$. Then following UDS, we would form the effective batch $\beta_i^{\text{eff}}$ by sharing data from $\beta_{j \neq i}$ if it falls above the $k^{th}$ percentile of Q values for $\beta_i$:

$$UDS_{\text{online}} : (s, a, r_i, s') \sim \beta_{j \neq i} \in \beta_i^{\text{eff}}$$
$$\text{if } \Delta^\pi(s, a) := \hat{Q}^\pi(s, a, i) - P_{k^{\text{th}}}[\hat{Q}^\pi(s', a', i) : s', a' \sim \beta_i] \geq 0$$

Note the differences here: (i) the 'data' used for data-sharing is the sampled replay buffer batch instead of the offline dataset, and (ii) we use the standard Q-function to evaluate data instead of the conservative Q-function since we are doing online (not offline) RL. We implement it this way as a practical approximation to avoid having to process the entire replay buffer every training iteration.

We use the same default hyperparameters as the other baseline methods. Additionally, we need to tune the sharing percentile $k$. For this, we tried $0^{\text{th}}$ percentile (sharing all data) and $80^{\text{th}}$ percentile, and chose the best-performing one.

