# OpenReview forum: "QMP: Q-switch Mixture of Policies for Multi-Task Behavior Sharing"
_ICLR.cc/2025/Conference — ICLR 2025 Poster_

### Official Review · Reviewer_CusU · 2024-10-22

**Soundness:** 2
**Presentation:** 3
**Contribution:** 2
**Rating:** 6
**Confidence:** 4

**Summary:**

This paper proposes a framework for multi-task reinforcement learning (MTRL) called Q-switch Mixture of Policies (QMP), which enables the selective sharing of behaviors between tasks to improve sample efficiency. QMP enhances off-policy data collection by selecting useful behaviors from other task policies based on the task's Q-function. The authors provide theoretical guarantees that QMP improves sample efficiency while preserving the convergence guarantees of reinforcement learning algorithms. Empirical evaluations demonstrate that QMP achieves complementary gains over existing MTRL algorithms in manipulation, locomotion, and navigation environments.

**Strengths:**

The paper is clearly written with a logical flow. The introduction and problem formulation effectively motivate the need for a new behavior-sharing method, and the diagrams provided help visualize the approach. The results are promising.

**Weaknesses:**

My impression of this paper is mixed. On the one hand, the authors demonstrate that QMP is effective. However, on the other hand, it's not intuitively clear why QMP works as well as it does. The proposed method is quite simple: it uses the current $Q_i$ function to select the argmax policy $\pi_j^{mix}$. Then, $\pi_j^{mix}$ (rather than $\pi_i$) is used to generate one-step data for training. I have some concerns about the approach of naively taking the maximum over per-task Q-functions—this may not be suitable in general settings, such as more stochastic environments, as it could lead to issues like switching tasks at every step or favoring certain tasks while neglecting others, which is undesirable. If this paper is accepted, it might be perceived as presenting a general framework for MTRL, but I have serious doubts about this claim.

In the introduction section of the paper, it mentions 'These tasks share many similar behaviors, like approaching the tabletop or grasping the object handle.' Therefore, I think 'behavior' refers to an action sequence or subtask that continues for a period of time. However, Algorithm 1 (one-step action sharing) appears to be misaligned with the paper's title and motivation, which emphasizes behavior (action sequence) sharing. The paper's motivation requires significant revision to address this discrepancy.

**Questions:**

Please refer to the Weaknesses.

---

> ### Author Response · Authors · 2024-11-21
> **Addressing questions on QMP mechanism and stability**
>
> We thank the reviewer for their detailed feedback and constructive comments. We are pleased that you found the paper well-written and the results promising. Below, we address your specific concerns point by point. We also made revisions to clarify the motivation and contributions of QMP based on your suggestions.
>
> ----
>
> > it's not intuitively clear why QMP works as well as it does. I have some concerns about the approach of naively taking the maximum over per-task Q-functions—this may not be suitable in general settings, such as more stochastic environments, as it could lead to issues like switching tasks at every step or favoring certain tasks while neglecting others, which is undesirable.
>
> We appreciate this observation and use Section 5 to clarify the mechanism why QMP works.
>
> ----
>
> ### Why QMP works:
> QMP works because at certain states, a task’s policy may lag behind its corresponding Q-function, failing to fully optimize the SAC objective:
> \\\[
> \\pi^{\\text{new}} \= \\arg \\min\_{\\pi' \\in \\Pi} D\_{\\text{KL}} \\left( \\pi'(\\cdot \\mid s\_t) \\,\\middle\\|\\, \\frac{\\exp\\left(\\frac{1}{\\alpha} Q^{\\pi^{\\text{old}}}(s\_t, \\cdot)\\right)}{Z^{\\pi^{\\text{old}}}(s\_t)} \\right)
> \\\]
>
> So, QMP leverages other task policies \\(\\pi\_j\\) that may have already learned optimal or near-optimal behaviors for similar states. By selecting the policy \\(\\pi\_j\\) that maximizes the above objective at a given state, QMP improves over the SAC objective, leading to greater policy improvement (Theorem 5.1) and the sample efficiency gains observed empirically.
>
> ----
>
> ### Why QMP is Stable:
> We would like to clarify that QMP does not “switch tasks at every step”. In one episode, the task stays fixed and the data is collected with its associated $\\pi\_i^{mix}$. At a given state, $\\pi\_i^{mix}$ selects the particular action-distribution $p(a) \= \\pi\_j(a|s)$ that collects data. Finally, the gathered data is all used to train $\\pi\_i$ only. Each task and policy receives equal episodes of data collection and gradient updates. This design avoids favoring certain tasks while maintaining stability (see Algorithm 1 for a formal description).
>
> ----
>
> ### Why naive maximization in Eq. (3) is sound, even in stochastic environments
> We understand it may be counter-intuitive, but the naive maximization step follows exactly from the policy improvement theorem, and is exactly what SAC optimizes. Appendix C derives the maximization step directly from the definition of soft policy improvement in SAC.
>
> The maximization step is robust in stochastic environments, because it is executed at every state and the Q-function accounts for future stochasticity. Concretely, at state $s\_k$, the selection is governed by $\\mathbb{E}\_\pi [Q(s\_k, a)]$. Under stochastic dynamics, the next state is $s\_{k+1} \\sim \\mathcal{T}(s\_k, a) $. Here, the selection will be governed by $\\mathbb{E}\_\pi [Q(s\_{k+1}, a)]$. So, since QMP is evaluated at every state, it ensures that QMP's mechanism is robust to stochastic environments.

---

> > ### Author Response · Authors · 2024-11-21
> > **Incorporated writing revisions**
> >
> > ### Revised generality of QMP to only off-policy MTRL
> >
> > > it might be perceived as presenting a general framework for MTRL, but I have serious doubts about this claim.
> >
> > We appreciate the reviewer for pointing this out and have revised the manuscript to explicitly note that QMP requires an off-policy RL algorithm since QMP uses $\\pi\_i^{mix}$ to gather data for training $\\pi\_i$. We have edited the writing to make this clear.
> >
> > 1. Introduction: *“This approach offers a simple, general, and sample-efficient approach that complements existing **off-policy** MTRL methods.”*
> > 2. Limitation: *“Since QMP only shares behaviors via **off-policy** data collection, it is not applicable to on-policy RL base algorithms like PPO.”*
> >
> > ----
> >
> > ### Revised motivation paragraph to explicitly define “behavior” as *how an agent acts in a particular state*
> >
> > > In the introduction section of the paper, it mentions 'These tasks share many similar behaviors, like approaching the tabletop or grasping the object handle.' Therefore, I think 'behavior' refers to an action sequence or subtask that continues for a period of time. However, Algorithm 1 (one-step action sharing) appears to be misaligned with the paper's title and motivation, which emphasizes behavior (action sequence) sharing. The paper's motivation requires significant revision to address this discrepancy.
> >
> > We thank the reviewer for pointing out this potential confusion. We agree that “behavior” could be interpreted as action sequences, which could cause confusion later in our paper, and we have revised the motivating paragraph to clarify our definition of behavior.
> >
> > **Definition**: In our paper, “behavior” refers to how an agent acts (action $a$) in response to a particular situation (state $s$), i.e., its policy $\\pi(a|s)$ at a given state. This definition aligns with established terminology in reinforcement and imitation learning:
> >
> > \[1\] Wikipedia, https://en.wikipedia.org/wiki/Behavior : "**behavior** is the internally coordinated responses (actions or inactions) of whole living organisms (individuals or groups) to internal or external stimuli"
> > \[2\] Richard S Sutton and Andrew G Barto. Reinforcement learning: An introduction. MIT press, 2018: **​​**"the **behavior** policy...is the policy controlling the agent and generating behavior."
> > \[3\] Stork, Jörg, et al. 2020, Understanding the behavior of reinforcement learning agents: “The **behavior of an RL agent** encompasses its (re-)actions, based on its environment and observed input states”
> > \[4\] Behavior cloning: behavior refers to expert decision-making, i.e., $\\pi(a|s)$ and behavior cloning means one-step cloning of the expert behavior per-step as opposed to sequence imitation $\\pi(a\_1, a\_2, a\_3, ..., a\_k | s)$
> >
> > **Behavior sharing definition**: Thus, “behavior sharing” simply means sharing of strategy $\\pi\_i(a|s)$ with another task $\\pi\_j(a|s)$ at a particular state $s$, so the behavior learned in one task, i.e. $\\pi\_i(a|s)$ is potentially useful for the other task.  We believe this definition of “behavior” is valid and commonly accepted.
> >
> > We appreciate your concern and have revised the motivating paragraph to define "behavior" explicitly in the rebuttal revision:
> >
> > *"In multi-task reinforcement learning, each task can benefit from the behaviors learned in others. Consider a robot learning four tasks simultaneously: opening and closing both a drawer and a door on a tabletop, as illustrated in Figure 1\. A behavior is defined as the policy of how the robot acts in response to a situation, with the optimal behavior representing the best response, such as opening its gripper (action) when near the drawer handle (state) in the drawer-open task. As the robot learns, such behaviors are often shareable between tasks. For instance, both drawer-open and drawer-close tasks require behaviors for grasping the handle. Consequently, as the robot refines its ability to grasp the drawer handle in one task, it can incorporate these behaviors into the other, reducing the need to explore the entire action space randomly. Following this intuition, can we develop a general framework that leverages such behavior sharing across tasks to accelerate overall learning?"*
> >
> > ----
> >
> > We thank you for your feedback and hope we've addressed your comments adequately. We are happy to answer any further questions and incorporate any further suggestions.

---

> > > ### Author Response · Authors · 2024-11-28
> > > **Thank you for your valuable time**
> > >
> > > We thank you for your valuable time and suggestions. We hope our rebuttal helps clarify the theoretically-backed simplicity of QMP and its promising results, and kindly request to reconsider your score if your concerns are sufficiently addressed. We would love the opportunity to discuss and incorporate any further suggestions to help make our paper more intuitive.
> > >
> > >
> > > ----
> > > &nbsp;
> > >
> > > To summarize,
> > > - We note that (i) there is no task switching — only one task per episode with different data collectors (Alg 1), and (ii) QMP works so well and is stable because its $\arg \max$ **theoretically** maximizes SAC’s objective better than SAC can do with only gradient ascent (Theorem 5.1).
> > > - We incorporated writing suggestions: (i) the motivation example now defines behavior explicitly, and (ii) intro and limitation emphasize “off-policy” MTRL.

---

> > > > ### Author Response · Authors · 2024-12-02
> > > >
> > > > Dear reviewer, we appreciate your time and effort so far to review our work. Since it is the last day of discussion, we would be grateful for any comments or suggestions in response to our rebuttal.
> > > >
> > > > If any part of our response requires further clarification, please let us know and we would be happy to incorporate any specific suggestions. If our response sufficiently addresses your concerns, we kindly ask you to consider revising your score. Thank you.

---

> > > > > ### Comment · Reviewer_CusU · 2024-12-03
> > > > >
> > > > > Thank you for your detailed response. I have updated my rating.

---

### Official Review · Reviewer_7Cb7 · 2024-10-31

**Soundness:** 2
**Presentation:** 3
**Contribution:** 2
**Rating:** 6
**Confidence:** 4

**Summary:**

The paper introduces QMP (Q-Switch Mixture of Policies), a novel framework for multi-task reinforcement learning (MTRL) designed to improve sample efficiency by selectively sharing behavioral policies across different tasks. Unlike traditional MTRL methods, which rely on parameter sharing or uniform behavior sharing, QMP identifies and shares beneficial behaviors from other tasks based on a Q-function evaluation. This approach enables each task to benefit from the learning progress in other tasks without introducing bias, as only helpful behaviors are integrated into the data collection phase.

**Strengths:**

1. The QMP framework introduces a Q-value-based behavior selection mechanism that enables selective behavior sharing in multi-task reinforcement learning (MTRL), enhancing sample efficiency.
2. The extensive experimental results across diverse domains—manipulation, navigation, and locomotion—demonstrate QMP’s performance compared to traditional behavior-sharing baselines, showcasing its practical impact in different tasks.

**Weaknesses:**

1. This paper only proposes a multi-task data sampling strategy implemented within an off-policy framework, which is limited in scope and shows limited improvement in more complex tasks, such as MT10 and MT50.
2. The other algorithms in paper [1] achieved better results with fewer interactions with the environment during training. I think that the method proposed in this paper makes a limited contribution to the field of multi-task reinforcement learning.
3. It is unclear how to ensure avoidance of local optima during complex multi-task transfer processes.
4. There is a lack of comparison with other MTRL baselines.

[1] Sodhani et al., 2021, Multi-Task Reinforcement Learning with Context-based Representations

**Questions:**

Please see previous section.

---

> ### Author Response · Authors · 2024-11-21
> **Addressing question on scope and performance on complex tasks**
>
> We thank the reviewer for their insightful feedback and constructive criticisms, and we address your concerns point by point below.
>
> As an overarching comment, our stated primary contributions are: introducing behavior policy sharing as a new avenue of information sharing in MTRL, proposing QMP as a simple and principled method for this purpose, and demonstrating its complementary gains to various MTRL settings. We do not claim QMP is a standalone state-of-the-art MTRL method but rather a plug-and-play component that can be integrated with existing MTRL frameworks, including parameter sharing methods like Sodhani et al. \[1\]. To this end, we hope our experiments sufficiently support our claims, showing complementary gains to existing MTRL and comparisons with baseline behavior-sharing methods.
>
> ----
>
> ### 1. Clarify scope and compare information sharing strategies
>
> > 1\. This paper only proposes a multi-task data sampling strategy implemented within an off-policy framework, which is limited in scope and shows limited improvement in more complex tasks, such as MT10 and MT50.
>
> We agree that QMP is a multi-task data sampling strategy by design: QMP is a simple, off-policy, and hyperparameter-free plug-in that complements other MTRL methods. We explicitly clarify the scope in the paper, and show how the improvements are significant, despite the simplicity of QMP.
>
> **Scope**: We have edited our paper to incorporate **off-policy** more explicitly, based on your feedback:
>
> 1. Introduction: “This approach offers a simple, general, and sample-efficient approach that complements existing **off-policy** MTRL methods.”
> 2. Limitation: “Since QMP only shares behaviors via **off-policy** data collection, it is not applicable to on-policy RL base algorithms like PPO.”
>
> **Improvements:** To provide additional perspective on performance across many benchmarks, the table below compares how much prior MTRL approaches improve the final performance over the simplest Zero-MTRL baseline, which is the dotted-blue line in Figure 7\.
>
> | MTRL approach v/s Zero-MTRL | Reacher | Maze | MT-CDS| Kitchen | MT10 |
> |:----------------|:--------------------:|:--------------------:|:------------------:|:--------------------:|:-----------------:|
> | Parameter-sharing (dotted-pink vs. dotted-blue) | \-35% | **\+50%** | **\+5%** | \+20% | \-5% |
> | Unsupervised data-sharing (dotted-green v/s dotted-blue) | \-50% | \-5% | \-10% | \+25% | **\+15%** |
> | QMP: Behavior-sharing (Solid blue v/s dotted-blue) | **\+25%** | \+15% | 0% | **\+50%** | **\+15%** |
>
> Two key insights emerge:
>
> - Behavior-sharing always results in non-negative gains across environments.
> - Behavior-sharing is competitive with and often outperforms other sharing methods.
>
> Moreover, in the **more complex MT50** task set, we observe an improvement of **\+22.8% over the parameter-sharing baseline**.

---

> > ### Author Response · Authors · 2024-11-21
> > **Address comparison with algorithms in Sodhani et al. [1]**
> >
> > ### 2. Compare with algorithms in Sodhani et al. \[1\]
> >
> > > 2\. The other algorithms in paper \[1\] achieved better results with fewer interactions with the environment during training. I think that the method proposed in this paper makes a limited contribution to the field of multi-task reinforcement learning.
> >
> > We respect the contribution of \[1\] and emphasize that algorithms in \[1\] have only been evaluated on MT10 and MT50 benchmarks, as opposed to our paper evaluating on 7 benchmarks. Based on your feedback, we give a fair comparison below.
> >
> > **CARE Algorithm \[1\]** requires privileged task metadata such as language description, and is fully complementary to QMP. While CARE is able to successfully use external metadata, it limits CARE’s applicability to benchmarks like Multistage Reacher, Maze Navigation, and Walker, where task metadata is not readily available.
> >
> > ----
> >
> > **Other algorithms in \[1\]** can be categorized in two ways, and their performance can be ranked based on MT10 (Table 1\) and MT50 (Table 3\) results from \[1\]:
> >
> > 1. Parameter-sharing: Multi-task SAC \< Multi-task SAC \+ Task Encoder \< **Multi-headed SAC**
> > 2. Gradient interference reduction:
> >    SAC \+ FiLM \< Soft Modularization $\\approx$ **PCGrad \[2\]**
> >
> >
> > Our paper already includes results on the best methods in these categories.
> >
> > **1\. Parameter-sharing with Multi-headed SAC:** Our results already demonstrate improvement over Multi-headed SAC in Figure 7 on 6 environments:
> >
> > | Approach | Reacher | Maze | MT-CDS| Kitchen | MT10 | MT 50 |
> > |:----------------|:--------------------:|:--------------------:|:------------------:|:--------------------:|:-----------------:|:-----------------:|
> > | Parameter-sharing (dotted-pink) | 0.55 | 0.90 | 0.95 | 0.48 | **0.7** | 0.35 |
> > | QMP \+ Parameter-sharing (solid pink) | **0.80** | **1.00** | **0.95** | **0.60** | **0.7** | **0.43** |
> >
> > **2\. Gradient interference reduction with PCGrad**: Our results on PCGrad in Figure 15 already demonstrate that QMP is complementary and improves over PCGrad:
> >
> > | Approach | Reacher | Maze | Kitchen | MT 50 |
> > |:----------------|:--------------------:|:--------------------:|:------------------:|:--------------------:|
> > | PCGrad (dotted-orange) | **0.78** | 0.90 | 0.55 | 0.35 |
> > | QMP \+ PCGrad (orange) | **0.78** | **1.00** | **0.60** | **0.42** |
> >
> > Key Takeaway: QMP improves performance over \[1\] when integrated with their baselines, achieving better results with no additional hyperparameter tuning.
> >
> > ----
> >
> > We note that the numbers are not directly comparable between ours and \[1\], likely due to implementation differences. As can be seen in the supplementary code, we have based our implementation on [https://github.com/rlworkgroup/garage](https://github.com/rlworkgroup/garage), which is a popular reinforcement learning toolkit. QMP shares the exact hyperparameters with Multi-head SAC and PCGrad in the results above.
> >
> > Nevertheless, for a fairer comparison, we normalize for the difference in implementation and try to compare the numbers with \[1\] below:
> >
> > **Timestep-normalized Comparison with algorithms in \[1\]**
> >
> > | Approach | PCGrad \[1\] | PCGrad (Ours) | QMP \+ PCGrad |
> > |:----------------|:--------------------:|:--------------------:|:------------------:|
> > | Performance | 0.21 | 0.21 | **0.31** |
> > | Reporting Timesteps | 5M | 5M | 5M |
> >
> > | Approach | MH-SAC \[1\] | MH-SAC (Ours) | QMP \+ MH-SAC |
> > |:----------------|:--------------------:|:--------------------:|:------------------:|
> > | Performance | 0.19 | 0.19 | **0.27** |
> > | Reporting Timesteps | 5M | 14M | 14M |
> >
> > Thus, our PCGrad implementation reaches the same success rate of 0.21 as \[1\] at 5M steps, and QMP \+ PCGrad is able to improve it to 0.31 at 5M steps. For MH-SAC, our implementation reaches 0.19 (that \[1\] reported at 5M) in 14M steps, and QMP \+ MH-SAC improves it to 0.27.

---

> > > ### Author Response · Authors · 2024-11-21
> > > **Addressing questions on local optima and MTRL baselines**
> > >
> > > 3. ### PCGrad addresses gradient interference
> > >
> > > > 3. It is unclear how to ensure avoidance of local optima during complex multi-task transfer processes.
> > >
> > > Thank you for your insightful comment and we agree this is an important consideration. While QMP only modifies the off-policy data collection, optimization challenges in MTRL, such as gradient interference, are addressed by methods like PCGrad \[2\]. As discussed above and in Figure 15, PCGrad improves over Multi-head SAC by reducing gradient interference. Crucially, QMP increases PCGrad performance further without introducing any additional instability.
> > >
> > > ----
> > >
> > > 4. ### QMP is complementary to various MTRL categories
> > >
> > > > 4. There is a lack of comparison with other MTRL baselines.
> > >
> > > We appreciate your feedback and clarify that QMP is not a standalone state-of-the-art MTRL method but rather a plug-and-play component that can complement existing MTRL frameworks. While we cannot show complementary results to the vast MTRL literature, our work includes the following comparisons to various categories of MTRL methods that do not assume privileged metadata (e.g., Sodhani et al. \[1\]):
> > >
> > > 1. No-Sharing: Base RL methods like SAC \[3\]
> > > 2. Data-Sharing: CDS \[4\], UDS\[5\]
> > > 3. Parameter-Sharing: Multi-headed SAC \[6\], DECAF \[7\]
> > > 4. Direct behavior sharing (biased): Divide-and-Conquer (DnC) and DnC-reg-only \[8\]
> > > 5. Reducing gradient interference: PCGrad \[2\]
> > >
> > > To the best of our knowledge, we have not missed any competing baseline to QMP, however, there are definitely a vast number of MTRL works which would be complementary to QMP. We have shown this on 4 different varieties of MTRL, to justify the utility and simplicity of QMP. On your recommendation, we are happy to include further discussion on any relevant MTRL work we might have missed.
> > >
> > > We hope the current comparisons highlight the key contributions of our work:
> > > - Introduce behavior-sharing as a novel, complementary avenue to improve off-policy MTRL sample efficiency.
> > > - QMP is complementary to various MTRL approaches like no-sharing, data-sharing, parameter-sharing, PCGrad (Figure 7, Figure 15\)
> > > - We are the first to propose “behavior-sharing **via off-policy data**”, and to validate this, we have compared to possible baselines on behavior-sharing like DnC and full-sharing (Figure 8\)
> > >
> > > \[1\] Sodhani et al., 2021, Multi-Task Reinforcement Learning with Context-based Representations.
> > > \[2\] Yu et al. 2020, Gradient surgery for multi-task learning.
> > > \[3\] Haarnoja et al. 2018, Soft actor-critic: Off-policy maximum entropy deep reinforcement learning with a stochastic actor.
> > > \[4\] Yu et al. 2020, Conservative data sharing for multi-task offline reinforcement learning.
> > > \[5\] Yu et al. 2022, How to leverage unlabeled data in offline reinforcement learning.
> > > \[6\] Mysore et al. 2022, Multi-critic actor learning: Teaching rl policies to act with style.
> > > \[7\] Glatt et al. 2020, Decaf: Deep case-based policy inference for knowledge transfer in reinforcement learning.
> > > \[8\] Ghosh et al. 2018, Divide-and-conquer reinforcement learning.
> > >
> > > ----
> > >
> > > We thank you for your feedback and hope we've addressed your comments adequately. We are happy to answer any further questions and incorporate any further suggestions.

---

> ### Author Response · Authors · 2024-11-28
> **Thank you for your valuable time**
>
> We appreciate your valuable time and we hope our rebuttal justifies the complementary benefits of a simple add-on QMP method across various environments and off-policy MTRL approaches. We would love to incorporate any further specific suggestions or concerns you have.
>
>
> &nbsp;
>
> ----
>
> In summary:
> 1. We revised “off-policy” MTRL as scope (Sec 1, 8)
> 2. QMP shows improvements on best algorithms from Sodhani et al [1] on MT10, MT50 and all tested environments (Fig 7, 15)
> 3. PCGrad addresses gradient interference, and PCGrad + QMP > PCGrad-only (Figure 15)
> 4. QMP is complementary to a wide coverage of MTRL baselines (Section 7).

---

> > ### Author Response · Authors · 2024-12-02
> >
> > Dear reviewer, we appreciate your time and effort so far to review our work. Since it is the last day of discussion, we would be grateful for any comments or suggestions in response to our rebuttal.
> >
> > If any part of our response requires further clarification, please let us know and we would be happy to incorporate any specific suggestions. If our response sufficiently addresses your concerns, we kindly ask you to consider revising your score. Thank you.

---

> > > ### Comment · Reviewer_7Cb7 · 2024-12-03
> > >
> > > Thank you to the authors for their effort in addressing the concerns and clarifying the questions. The authors' response addressed most of my concerns, and I have adjusted my score accordingly.

---

### Official Review · Reviewer_ENcC · 2024-11-03

**Soundness:** 3
**Presentation:** 3
**Contribution:** 3
**Rating:** 8
**Confidence:** 4

**Summary:**

This paper introduces the Q-switch Mixture of Policies (QMP), a framework for multi-task reinforcement learning (MTRL) aimed at enhancing sample efficiency by sharing behaviors selectively across tasks. The approach enables an agent to leverage useful behaviors from other tasks during off-policy data collection, guided by the Q-function of the current task to identify beneficial actions. Unlike traditional MTRL methods that rely on uniform sharing or regularization, QMP avoids biasing policy objectives by only sharing behaviors during data collection rather than directly influencing policy updates. Experimental results across various environments, including manipulation and locomotion tasks, demonstrate QMP's capability to accelerate training and improve performance when integrated with existing MTRL frameworks.

**Strengths:**

1. QMP is designed to complement existing MTRL frameworks, such as parameter sharing and data relabeling.
2. The authors provide theoretical analysis showing that QMP’s behavior-sharing mechanism preserves the convergence guarantees of the underlying reinforcement learning algorithm.
3. The paper presents extensive experiments across various multi-task environments (e.g., manipulation, navigation, and locomotion).

**Weaknesses:**

1. QMP requires evaluating Q-values across multiple task policies at each decision step, which could introduce computational overhead, particularly in settings with a large number of tasks. The paper lacks a detailed comparison of computational costs between QMP and baseline methods, which would be helpful for assessing its scalability.

**Questions:**

1. Could you provide some information regarding the computational overhead of QMP?

---

> ### Author Response · Authors · 2024-11-21
> **Addressing question on compute time and scalability**
>
> We thank the reviewer for their thoughtful feedback and positive evaluation of our work. We are glad that you appreciated QMP’s complementary design, theoretical soundness, and extensive experiments. Below, we address your question regarding computational overhead and scalability.
>
> > QMP requires evaluating Q-values across multiple task policies at each decision step, which could introduce computational overhead, particularly in settings with a large number of tasks. The paper lacks a detailed comparison of computational costs between QMP and baseline methods, which would be helpful for assessing its scalability.
>
> Thank you for your question.  Appendix H.4 and Table 3 in the paper provide a detailed computational cost analysis. To summarize:
>
> ### 1. Small overhead in computational time
>
> As shown in Table 3, QMP introduces only a **1.75% time overhead** on the large 50-task set in MT50, because the total runtime is dominated by the compute cost of neural network training and environment simulation speed.
>
> | Environment           | No-Sharing | QMP \+ No-Sharing | Parameter-Sharing | QMP \+ Parameter-Sharing |
> |-----------------------|------------|-------------------|--------------------|--------------------------|
> | Reacher Multistage    | 12.5 hr    | 14.2 hr          | 14 hr             | 16.2 hr                 |
> | MT50                 | \-          | \-                 | 7 days, 3 hr      | 7 days, 6 hr            |
>
> ----
>
> ### 2. Efficient policy and Q-function evaluation
>
> QMP requires evaluating N policy proposals and N Q-values to sample from \\( \\pi^i\_{mix} \\). However, these evaluations are efficiently **batched and parallelized**.
> \- In QMP \+ Parameter-Sharing, the multihead architectures of the policy and Q-network allow all N evaluations to be performed in one single pass through each network. Among these two, the policy pass is already performed to get the action from the current task’s policy, so the only extra overhead is due to one Q-network pass.
> \- Without parameter-sharing, \\( Q\_i(s, a\_j) \\) evaluations can be batched over the number of actions, and the policy evaluations \\( \\pi\_j(a | s) \\), being independent, can be obtained in parallel. Our implementation batched the Q evaluations, but did not parallelize the policy evaluations, thus incurring a small computational overhead of around 2 hr.
>
> ----
>
> ### 3. Scalability
> Due to the parallelizable nature of QMP’s operations, its overhead scales well even for large task sets and in combination with parameter-sharing, requiring the extra time for **only a single Q-function pass**.
>
> ----
>
> We thank you for your feedback and hope we've addressed your comments adequately. We are happy to answer any further questions and incorporate any further suggestions.

---

> ### Author Response · Authors · 2024-12-02
>
> Dear reviewer, we appreciate your time and effort to review our work. Since it is the last day of discussion, we would be grateful for any comments or suggestions in response to our rebuttal. We would be happy to incorporate any specific suggestions. Thank you.

---

### Official Review · Reviewer_MdZX · 2024-11-03

**Soundness:** 2
**Presentation:** 3
**Contribution:** 3
**Rating:** 6
**Confidence:** 3

**Summary:**

This paper introduces Q-switch Mixture of Policies (QMP), a novel framework for multi-task reinforcement learning that improves sample efficiency through selective behavior sharing between tasks. The key idea is to enhance each task's off-policy data collection by selectively employing behaviors from other task policies, using the task's Q-function to evaluate and select useful shareable behaviors.

**Strengths:**

- The paper is well-written with clear organization and easy to follow.

- The method is elegantly simple yet theoretically sound, introducing behavior sharing through off-policy data collection rather than policy regularization, which avoids bias in the learning objective while maintaining convergence guarantees.

- The proposed Q-switch mechanism provides a principled approach to selective behavior sharing that is complementary to existing MTRL methods, demonstrating consistent improvements when combined with parameter sharing, data sharing, and gradient-based approaches.

**Weaknesses:**

1. The Q-switch mechanism requires evaluating all task policies and Q-values at each step. While parallelization helps, this still leads to significant computational costs, especially for large task sets as evidenced by the 7+ days runtime in MT50 experiments.
2. The method heavily relies on accurate Q-function estimation to select appropriate behaviors. This dependency may lead to suboptimal behavior selection during early training or in tasks with sparse rewards where Q-function learning is unstable.

3. The method requires careful tuning of hyperparameters, such as manually setting 70% task-specific policy usage in Meta-World MT10. This tuning requirement may limit practical applicability.

4. The method's effectiveness heavily depends on the existence of shareable behaviors between tasks. Performance gains might be limited for completely unrelated or highly conflicting task sets.

5. The convergence analysis primarily focuses on tabular MDPs. The theoretical guarantees may not fully extend to continuous state-action spaces with function approximation.

**Questions:**

See above.

---

> ### Author Response · Authors · 2024-11-21
> **Addressing questions on compute, Q-function accuracy, one hyperparameter in MT10.**
>
> We sincerely thank the reviewer for their thoughtful evaluation and valuable feedback. We are pleased that you appreciated the elegance and theoretical soundness of our QMP framework, as well as its consistently complementary nature to existing MTRL methods. We address your concerns below.
>
> ### 1. QMP adds negligible compute time
>
> > significant computational costs, especially for large task sets as evidenced by the 7+ days runtime in MT50 experiments
>
> We respectfully clarify that this is a misunderstanding. Table 3 shows that Parameter-Sharing baseline (no QMP) on MT50 already takes 7 days, 3 hr, while QMP \+ Parameter-Sharing takes 7 days, 6 hr — only a **1.75% time overhead** on the large 50-task set. The total runtime is dominated by the compute cost of neural network training and environment simulation speed. In comparison, the additional compute time due to QMP is nominal. With parallelization, only one additional Q-function evaluation per policy step is needed.
>
> ----
>
> ### 2. QMP does not introduce additional dependence on accurate Q-function estimation
>
> > The method heavily relies on accurate Q-function estimation to select appropriate behaviors.
>
> This concern likely stems from the explicit $\\arg \\max$ step before selecting a policy for action. However, the QMP objective follows directly from SAC’s underlying objective, and both are equally dependent on accurate Q-function estimation:
> \\\[
> \\pi^{\\text{new}} \= \\arg \\min\_{\\pi' \\in \\Pi} \\mathrm{D}\_{\\mathrm{KL}} \\left( \\pi'(\\cdot \\mid s\_t) \\Bigg\\| \\frac{\\exp \\left(\\frac{1}{\\alpha}Q^{\\pi^{\\text{old}}}(s\_t, \\cdot)\\right)}{Z^{\\pi^{\\text{old}}}(s\_t)} \\right)
> \\\]
>
> While SAC performs gradient ascent over the current Q-function (which could be inaccurate), QMP layers an additional maximization step on top. Thus, the explicit $\\arg \\max$ in QMP is simply achieving what gradient ascent aims to do for SAC, just better — i.e., maximize whatever the current Q-function is. Both rely equally on the Q-function and do not alter their dependence on Q-function accuracy. These algorithms still work because of how generalized policy iteration (GPI) (see Section 5.1 and below).
>
> **Generalized Policy Iteration (GPI):** GPI converges even with initially inaccurate Q-functions, as long as iterative updates are made. As per Sutton and Barto \[1\], in GPI, “policy evaluation and policy improvement processes interact, independent of the granularity and other details of the two processes.” Theorem D.2 shows that GPI with QMP converges at least as fast as GPI without QMP, which means that no additional instability is introduced due to QMP.
>
> > This dependency may lead to suboptimal behavior selection during early training or in tasks with sparse rewards where Q-function learning is unstable.
>
> **Sparse Rewards:** It is true that sparse rewards delay (soft) Q-function learning, but this affects QMP and SAC equally, wherein the actor learning would be delayed because the primary source of actor updates is gradient ascent over the Q-function. Moreover, QMP accelerates learning once Q-function updates become positive by leveraging behaviors from other tasks more effectively. This is evident in Task 3 of Multistage Reacher (Figure 13), a sparse reward task, where QMP achieves the most improvement as it shortcuts the delay in learning of its own actor, by using other task policies.
>
> ----
>
> ### 3. All tasks except MT10 were hyperparameter-free
>
> > The method requires careful tuning of hyperparameters, such as manually setting 70% task-specific policy usage in Meta-World MT10.
>
> This is true for MT10, where 70% conservative behavior-sharing led to minor improvements. With both 50% and 70%, we observed similar performance gains and did not tune this single hyperparameter further. In all other environments, conservative behavior-sharing was unnecessary, and no hyperparameter tuning was needed, and we observed this peculiar behavior only in MT10 and reported it transparently in the paper. Notably, the larger MT50 set result does not have any hyperparameter tuning.

---

> > ### Author Response · Authors · 2024-11-21
> > **Addressing questions on conflicting task sets, convergence results.**
> >
> > ### 4. QMP leverages shareable behaviors, while being robust to conflicting behaviors
> >
> > > The method's effectiveness heavily depends on the existence of shareable behaviors between tasks. Performance gains might be limited for completely unrelated or highly conflicting task sets.
> >
> > This is accurate and aligns with our conclusion: “QMP does not assume that optimal task behaviors always coincide. Thus, its improvement magnitude is limited by the degree of shareable behaviors and the suboptimality gap that exists.”
> >
> > QMP, being a behavior-sharing method, would only benefit multi-task RL if there are helpful behaviors to share across tasks. Importantly, it stays robust for task sets with unrelated or conflicting behaviors, as Theorem D.2 shows. In contrast, other methods can degrade performance in such scenarios as they bias the policy while behavior-sharing (Figure 8).
> >
> > For tasks with a mix of shareable and conflicting behaviors, QMP’s Q-switch selectively extracts shareable behaviors which leads to a better performance in the Multistage Reacher mixed task set (Figure 13).
> >
> > ----
> >
> > ### 5. Convergence theory cannot be extended to neural networks
> >
> > > The convergence analysis primarily focuses on tabular MDPs. The theoretical guarantees may not fully extend to continuous state-action spaces with function approximation.
> >
> > Respectfully, this is true for all deep reinforcement learning (DRL) algorithms built on neural networks. QMP’s theoretical guarantees are based on the same tabular assumptions as the most common continuous action space RL algorithms like SAC \[2\], DDPG \[3\], and TD3 \[4\], none of which extend function approximation guarantees. To quote DDPG paper,
> >
> > *“The use of large, non-linear function approximators for learning value or action-value functions has often been avoided in the past since theoretical performance guarantees are **impossible**”*
> >
> > Like standard practice in deep RL, we hope the convergence analysis offers value as a proof of soundness for the algorithm to back its empirical performance in multi-task RL.
> >
> > ----
> >
> > \[1\] Sutton & Barto. Reinforcement learning: An introduction. MIT press, 2018\.
> > \[2\] Tuomas Haarnoja et al. Soft actor-critic: Off-policy maximum entropy deep reinforcement learning with a stochastic actor.  ICML, 2018\.
> > \[3\] Timothy P. Lillicrap et al. Continuous control with deep reinforcement learning. ICLR, 2016\.
> > \[4\] Scott Fujimoto et al. Addressing Function Approximation Error in Actor-Critic Methods. ICML 2018\.
> >
> > ----
> >
> > We thank you for your feedback and hope we've addressed your comments adequately. We are happy to answer any further questions and incorporate any further suggestions.

---

> ### Author Response · Authors · 2024-11-28
> **Thank you for your valuable time**
>
> We thank you for your valuable time. In summary, we show:
> 1. compute overhead is low (Table 3)
> 2. QMP is reliable because of GPI (Sec 5.1)
> 3. no hyperparameter-tuning is required (Appendix I)
> 4. QMP especially outperforms baselines in conflicting tasks (Figure 13).
>
> &nbsp;
>
> If we have addressed all your concerns and justified the simplicity and soundness of QMP, we kindly request you to reconsider your rating. Please let us know if we can address any specific concerns or suggestions, as we would love the opportunity to discuss.

---

> > ### Author Response · Authors · 2024-12-02
> >
> > Dear reviewer, we appreciate your time and effort so far to review our work. Since it is the last day of discussion, we would be grateful for any comments or suggestions in response to our rebuttal.
> >
> > If any part of our response requires further clarification, please let us know and we would be happy to incorporate any specific suggestions. If our response sufficiently addresses your concerns, we kindly ask you to consider revising your score. Thank you.

---

> ### Comment · Reviewer_MdZX · 2024-12-03
> **Thanks for the rebuttal**
>
> I appreciate the authors' thorough response to the previous concerns and their transparent discussion of these limitations. The revision has significantly improved the paper's clarity and completeness.
>
> While this is a fundamental limitation of behavior-sharing approaches rather than a weakness specific to QMP, it would be valuable for future work to quantify this relationship between behavioral similarity and expected performance gains.
>
> Overall, I will raise my score accordingly.

---

### Author Response · Authors · 2024-11-25
**Summary of rebuttal. Kindly requesting discussion or response.**

We would appreciate the opportunity to discuss any remaining concerns with the reviewers. Following the valuable feedback from the reviewers, we have made several revisions and clarifications to address their concerns and improve the clarity and impact of our work.

We are pleased that the reviewers appreciated the *clarity* of our writing, the *novelty* and *simplicity* of our approach, its *theoretical* soundness, and the *extensive experiments* across diverse domains, which show promising results demonstrating how QMP is *complementary* to a variety of MTRL methods.

----

## Summary of revisions and clarifications

----

### 1. Negligible compute overhead of QMP: +1.75% time for +22.8% performance in MT50 (Reviewer MdZX, ENcC)

We clarify that QMP scales really well because its operations are parallelizable. In 50-task Metworld, QMP adds only a **1.75% time overhead** on the baseline due to a single extra Q-function pass (Table 4), yielding a significant performance improvement of +22.8% for a minimal increase in computation time.

----

### 2. QMP’s maximization objective is stable and consistent with GPI and SAC  (Reviewer MdZX, CusU)
- We clarify that QMP’s $\arg \max$ step is not unnatural and is consistent with the *fundamental objective underlying SAC*, as maximization is inherent in SAC’s formulation (see Eq. 1). While **SAC maximizes via gradient ascent**, QMP enhances (not replace) this by an **explicit maximization** over available policies. Consequently, Theorem D.2 theoretically proves that QMP converges at least as fast as SAC.
- Importantly, QMP **does not switch tasks** in between — the MDP (transition dynamics and reward function) remains fixed in one episode. QMP only switches the best policy to collect data for the current task at each decision point.
- By optimizing the generalized policy iteration (GPI) objective more effectively than SAC, QMP actually **improves robustness** to *Q-function estimation* (by quickly exploring and correcting), *stochastic environments*, *sparse reward* (Task 3 in Figure 13), and *conflicting tasks* (task 4 in Figure 13), because QMP accelerates the policy improvement step inherent to GPI.

----

### 3. Writing Revision: Emphasize *off-policy* MTRL as QMP’s scope and explicitly define *behavior* in motivation (Reviewer 7Cb7, CusU)

- **Scope**: In introduction and limitation, we explicitly note that QMP is a simple, off-policy, and hyperparameter-free plug-in that complements existing off-policy MTRL by modifying the data collection, and thus, not applicable to on-policy RL algorithms like PPO.

- **Behavior definition**: We revised the motivating paragraph in the introduction to explicitly define 'behavior' as the policy of how an agent acts in response to a particular state, aligning with established terminology in reinforcement and imitation learning, and eliminating any doubts that behavior-sharing might mean “multi-step action sharing” or “temporally-extended behavior sharing” (which is a separate heuristic extension in Appendix F.6)

----

### 4. Comparison to MTRL baselines: QMP complements various MTRL frameworks (Reviewer 7Cb7)
- We show QMP’s behavior-sharing **improves the performance when combined** with 4 frameworks of MTRL, including the best-performing methods from Sodhani et al. [1]:
1. No-Sharing: Base RL methods like SAC [3]
2. Data-Sharing: CDS [4], UDS[5]
3. Parameter-Sharing: Multi-headed SAC [6], DECAF [7]
4. Reducing gradient interference to **avoid local optima**: PCGrad [2]

For instance, on the best approach PCGrad, QMP still improves further!
| Approach | Reacher | Maze | Kitchen | MT 50 |
|:------|:---------:|:----:|:------:|:------:|
| PCGrad (dotted-orange) | **0.78** | 0.90 | 0.55 | 0.35 |
| QMP \+ PCGrad (orange) | **0.78** | **1.00** | **0.60** | **0.42** |

---

- We also show that QMP’s off-policy behavior-sharing is better than **baseline** direct behavior sharing methods that bias each policy: Divide-and-Conquer [8] and full-sharing.

- Even as a standalone improvement, QMP’s gains are competitive to other ways to share information like via parameters or data. Crucially, QMP’s behavior-sharing gains are positive, orthogonal and complementary, so it is an easy-to-add and robust plug-in for off-policy MTRL.
| MTRL approach v/s Zero-MTRL | Reacher | Maze | MT-CDS| Kitchen | MT10 |
|:----|:-----------:|:-----:|:-----:|:--------:|:------:|
| Parameter-sharing (dotted-pink vs. dotted-blue) | \-35% | **\+50%** | **\+5%** | \+20% | \-5% |
| Unsupervised data-sharing (dotted-green v/s dotted-blue) | \-50% | \-5% | \-10% | \+25% | **\+15%** |
| QMP: Behavior-sharing (Solid blue v/s dotted-blue) | **\+25%** | \+15% | 0% | **\+50%** | **\+15%** |

[References in response to Reviewer 7Cb7]

----

We thank the reviewers for your valuable time and kindly request you to reconsider your scores if we have satisfactorily addressed your concerns. We are happy to incorporate any further feedback.

---

### Meta-Review · Area_Chair_zji7 · 2024-12-19

**Metareview:**

This paper introduces QMP (Q-switch Mixture of Policies), a novel framework for sharing behaviors across tasks in multi-task reinforcement learning through off-policy data collection. The reviewers appreciated the paper's clear writing and theoretical soundness, particularly in demonstrating how QMP complements existing MTRL methods. Through extensive revisions, the authors effectively addressed initial concerns about computational overhead (showing only 1.75% overhead for 22.8% performance gain), clarified the stability of QMP's maximization objective, and better positioned the work within the off-policy MTRL context. While some reviewers initially expressed concerns about QMP's generality beyond grid worlds and its reliance on Q-function accuracy, the authors provided convincing theoretical and empirical evidence for QMP's robustness. Moving forward, exploring QMP's behavior sharing benefits with more complex task relationships would be valuable.

Given the paper's solid theoretical foundation, thorough empirical validation across diverse environments, and demonstrated complementary gains to existing MTRL approaches, I recommend accepting this work.

**Additional Comments On Reviewer Discussion:**

The reviewers appreciated the paper's clear writing and theoretical soundness, particularly in demonstrating how QMP complements existing MTRL methods. Through extensive revisions, the authors effectively addressed initial concerns about computational overhead (showing only 1.75% overhead for 22.8% performance gain), clarified the stability of QMP's maximization objective, and better positioned the work within the off-policy MTRL context. While some reviewers initially expressed concerns about QMP's generality beyond grid worlds and its reliance on Q-function accuracy, the authors provided convincing theoretical and empirical evidence for QMP's robustness.

---

### Decision · Program_Chairs · 2025-01-22

Accept (Poster)